# Quantifying prevalence and risk factors of HIV multiple infection in Uganda from population-based deep-sequence data

**Michael A. Martin**[1], **Andrea Brizzi**[2], **Xiaoyue Xi**[2,3], **Ronald Moses Galiwango**[4], **Sikhulile Moyo**[5,6], **Deogratius Ssemwanga**[7,8], **Alexandra Blenkinsop**[2], **Andrew D. Redd**[9,10,11], **Lucie Abeler-Dörner**[12], **Christophe Fraser**[12], **Steven J. Reynolds**[4,9,10], **Thomas C. Quinn**[4,9,10], **Joseph Kagaayi**[4,13], **David Bonsall**[14], **David Serwadda**[4], **Gertrude Nakigozi**[4], **Godfrey Kigozi**[4], **M. Kate Grabowski**[1,4,15*], **Oliver Ratmann**[2*], **with the PANGEA-HIV Consortium and the Rakai Health Sciences Program**

1 Department of Pathology, Johns Hopkins School of Medicine, Baltimore, Maryland, United States of America, 2 Department of Mathematics, Imperial College London, London, United Kingdom, 3 Medical Research Council Biostatistics Unit, University of Cambridge, Cambridge, United Kingdom, 4 Rakai Health Sciences Program, Kalisizo, Uganda, 5 Botswana Harvard AIDS Institute Partnership, Botswana Harvard HIV Reference Laboratory, Gaborone, Botswana, 6 Harvard T.H. Chan School of Public Health, Boston, Massachusetts, United States of America, 7 Medical Research Council/Uganda Virus Research Institute and London School of Hygiene and Tropical Medicine Uganda Research Unit, Entebbe, Uganda, 8 Uganda Virus Research Institute, Entebbe, Uganda, 9 Department of Medicine, Johns Hopkins School of Medicine, Baltimore, Maryland, United States of America, 10 Division of Intramural Research, National Institute of Allergy and Infectious Diseases, National Institutes of Health, Bethesda, Maryland, United States of America, 11 Institute of Infectious Disease and Molecular Medicine, University of Cape Town, Cape Town, South Africa, 12 Pandemic Sciences Institute, Nuffield Department of Medicine, University of Oxford, Oxford, United Kingdom, 13 Makerere University School of Public Health, Kampala, Uganda, 14 Wellcome Centre for Human Genetics, Nuffield Department of Medicine, University of Oxford, Oxford, United Kingdom, 15 Department of Epidemiology, Johns Hopkins Bloomberg School of Public Health, Baltimore, Maryland, United States of America

* mmart108@jhmi.edu (MAM); mgrabow2@jhu.edu (MKG); oliver.ratmann@imperial.ac.uk (OR)

**Data availability statement:** Processed phyloscanner output, de-identified epidemiological metadata, and all analysis

## Abstract

People living with HIV can acquire secondary infections through a process called super-infection, giving rise to simultaneous infection with genetically distinct variants (multiple infection). Multiple infection provides the necessary conditions for the generation of novel recombinant forms of HIV and may worsen clinical outcomes and increase the rate of transmission to HIV seronegative sexual partners. To date, studies of HIV multiple infection have relied on insensitive bulk-sequencing, labor intensive single genome amplification protocols, or deep-sequencing of short genome regions. Here, we identified multiple infections in whole-genome or near whole-genome HIV RNA deep-sequence data generated from plasma samples of 2,029 people living with viremic HIV who participated in the population-based Rakai Community Cohort Study (RCCS). We estimated individual- and population-level probabilities of being multiply infected and assessed epidemiological risk factors using the novel Bayesian deep-phylogenetic multiple infection model ($deep-phyloMI$) which accounts for bias due to partial sequencing success and false-negative and false-positive detection rates. We estimated that between 2010 and 2020, 4.09% (95% highest posterior density interval (HPD) 2.95%–5.45%) of RCCS

and visualization code is available at https://github.com/m-a-martin/rccs_hiv_moi. Bayesian model fit diagnostics are also available on the GitHub repository. HIV consensus sequences are available from Zenodo (https://doi.org/10.5281/zenodo.10075814) and the PANGEA-HIV sequence repository (https://github.com/PANGEA-HIV/PANGEA-Sequences) as open-access dataset under the CC-BY-4.0 license. HIV-1 deep-sequence reads can be requested from PANGEA-HIV under a managed access policy due to privacy and ethical reasons, which aligns with UNAIDS ethical guidelines. The process for accessing data, the PANGEA-HIV data sharing policy and a detailed description of what data are available is described at https://www.pangea-hiv.org/join-us. For more information contact PANGEA at pangea.data.enquiries@ndm.ox.ac.uk. The time frame for a response to requests is 2–4 weeks. Additional cohort data can be requested from RHSP. Because of privacy and ethical reasons, RHSP maintains a controlled access data policy for corresponding epidemiological metadata and corresponding data collection tools. In brief, RHSP policy requires individuals to submit an RHSP data request form (available upon request from datarequests@rhsp.org) and a brief concept note (one or two pages) detailing their research questions and methods. In addition, researchers are asked to provide a curriculum vitae/resume along with proof of human subjects research training. The time frame for a response to requests is 2–4 weeks.

**Funding:** This study was supported by the Bill and Melinda Gates Foundation (https://www.gatesfoundation.org, OPP1084362, INV-007573, INV-035619, INV-060259, INV-075093 to D.S., S.M., C.F., M.K.G., and O.R.), the National Institute of Health (NIH) National Institute of Allergy and Infectious Diseases (NIAID, https://www.niaid.nih.gov, U01AI075115, R01AI087409, U01AI100031, R01AI110324, R01AI114438, K25AI114461, R01AI123002, K01AI125086 to M.K.G., R01AI128779, R01AI143333, R21AI145682, R01AI155080 to M.K.G., ZIAAI001040 to T.C.Q.), NIH National Institute of Child Health and Development (https://www.nichd.nih.gov, R01HD050180, R01HD070769, R01HD091003), NIH National Heart, Lung, and Blood Institute (https://www.nhlbi.nih.gov, R01HL152813), the Fogarty International Center (https://www.fic.nih.gov, D43TW009578, D43TW010557), the Johns Hopkins University Center for AIDS Research (https://hopkinscfar.org, P30AI094189), the U.S. President's

participants with viremic HIV multiple infection at time of sampling. Participants living in high-HIV prevalence communities along Lake Victoria were 2.33-fold (95% HPD 1.3–3.7) more likely to harbor a multiple infection compared to individuals in lower prevalence neighboring communities. This work introduces a high-throughput surveillance framework for identifying people with multiple HIV infections and quantifying population-level prevalence and risk factors of multiple infection for clinical and epidemiological investigations.

## Author summary

HIV exists as a population of genetically distinct viral variants among people living with HIV. People living with HIV can be infected with genetically distinct variants. Identification of these mixed infections requires generating viral genomic data from people living with HIV. In the past, the approaches used to identify multiple infections from viral genomic data have had poor sensitivity or required labor intensive protocols that are prohibitive in application to large data sets. Prior work has also only utilized data generated from small portions of the viral genome and the statistical procedures used to generate population-level estimates from sequencing data generated from individual infections has not accounted for incomplete sampling of the within-host viral population or sources of sequencing error, which may confound multiple infection estimates. Here, we develop a statistical model that addresses these limitations and allows for the identification of multiple infections and the estimation of population-level risk of multiple infection from deep-sequence data. We fit this model to population-based HIV genomic data from people living with HIV in southern Uganda and estimate that approximately 4% of viremic participants harbor a multiple infection at a given point in time. We show that the prevalence of multiple infections is higher in key populations with high HIV prevalence. These findings inform our understanding of the sexual risk networks that give rise to multiple infections and aid in efforts to model HIV epidemiological dynamics and evolution during a period of incidence declines and shifting transmission dynamics across Eastern and Southern Africa.

## 1. Introduction

Simultaneous infection with multiple distinct variants of human immunodeficiency virus (HIV) can occur through a process called superinfection following secondary exposure to infected bodily fluids [1]. Following acquisition, infecting variants are shaped by within-individual evolutionary processes and can either stably coexist or undergo competitive exclusion [2,3]. Superinfection of PLHIV has important implications for the evolution, pathogenesis, and spread of HIV. Specifically, it provides the necessary conditions for the generation of novel recombinant viruses [4,5], which fuels diversification of the circulating viral population [6,7], complicating vaccine development efforts through the generation of novel epitopes [8,9] and potentially leads to the evolution of more transmissible viral genotypes [10]. Acquisition of superinfections may also increase the breadth and strength of the antibody response to HIV infection [11–13], potentially aiding in the identification of broadly neutralizing antibodies [14]. Finally, multiple infections may themselves lead to faster disease progression [15–17] and higher viral load [16,17], thereby potentially also increasing the risk of onward transmission [18,19]. While the availability of viral genome sequence data has allowed

Emergency Plan for AIDS Relief through the Centers for Disease Control and Prevention (https://www.state.gov/pepfar/, NU2GGH000817), and in part by the Division of Intramural Research, NIAID, NIH (A.D.R., S.J.R., T.C.Q.). MM was supported in part by NIH 1L30AI178824. The funders had no role in study design, data collection and analysis, decision to publish, or preparation of the manuscript.

**Competing interests:** The authors have declared that no competing interests exist.

for the identification of HIV multiple infections across a range of epidemiological contexts [20], prevalence estimates have generally been based on relatively small samples sizes with only partial genome data. Here, we identify HIV multiple infections using within-host deep-sequence phylogenetic trees inferred across the genome from a population-based surveillance cohort.

To date, viral sequence-based methods to identify HIV multiple infections have generally relied on one of three approaches. First, bulk sequencing (e.g. Sanger sequencing or consensus sequence estimation from deep-sequence data) can reveal instances where the majority viral variant changes between baseline and follow-up visits under longitudinal sampling or cases where the within-person viral population at a specific visit harbors abnormal levels of diversity [16,21–23]. While this approach proved useful prior to the availability of deep-sequencing technologies, it has a sensitivity of only $\sim 5\%$ to detect variants present in $\leq 20\%$ of the viral population within a sample [24]. Alternatively, single genome amplification (SGA) relies on serial dilutions to isolate a single molecule of transcribed viral cDNA prior to amplification and sequencing [25–27]. This approach is more sensitive in detecting minor variants than bulk sequencing and was considered the "gold standard" [28], but is labor intensive and difficult to apply at scale. Amplicon deep-sequencing of discrete regions of the HIV genome is able to achieve high sensitivity while being highly scalable to large sets of samples and has therefore been broadly applied to study multiple infections in larger studies [2,28–31].

Despite advancements in viral sequence-based identification of HIV multiple infections, existing approaches share shortcomings that hinder the interpretation of the results they generate. Critically, all of these methods rely on sequence data generated from only a subset of the genome, due in part to historical challenges in generating whole-genome HIV sequence data. For example, general population-based studies in Rakai, Uganda have previously utilized sequence data from 390 base pairs (bp) and 324 bp of the p24 (*gag*) and gp41 (*env*) regions, representing only 7.3% of the HIV genome. This inherently limits sensitivity to identify multiple infection with viral variants that are highly related within these short regions. Analysis of *gag* sequence data sampled from high-risk Kenyan women revealed cases of superinfection that were unidentified when querying only the *env* region [32]. Further, limited consideration has been given to the fact that factors that affect sequencing success of biological samples [33] may also affect the detection probability of multiple viral variants and may therefore confound prevalence estimates and assessment of multiple infection risk factors. Finally, existing methods generally use binary categorization of samples as either multiply or singly infected. They do not quantify uncertainty in individual-level assignments and do not account for this uncertainty when estimating population-level prevalence. With the advent of approaches that can generate near whole-genome HIV deep-sequence data [33,34], there is a need for statistical approaches that can integrate data from across the genome to robustly identify multiple infections while accounting for the various sources of bias that can obscure the underlying biological signal.

Here, we identify individuals that are likely to have multiple HIV multiple infection at the time of sampling, provide minimum estimates of the prevalence of HIV multiple infections in Rakai, Uganda between January 2010 and November 2020, and characterize risk factors for harboring a multiple infection based on HIV RNA deep-sequence data obtained from plasma samples of 2,029 people living with viremic HIV aged 15-49 who participated in the longitudinal, population-based Rakai Community Cohort Study (RCCS) [35,36]. These estimates reflect multiple infections present at time of sampling in plasma and, because infecting variants may be lost over time due to within-host evolutionary processes [2,3], should be interpreted as the minimum prevalence of people who have ever been multiply infected. Rakai District is located in south-central Uganda, East Africa, bordering Lake Victoria, and

is one of the areas with highest HIV-prevalence globally [37]. To support these inferences, we developed a novel Bayesian statistical model to identify multiple infections using within-host phylogenetic trees inferred from deep-sequence data generated from across the HIV genome, which we call the deep-phylo multiple infection model (*deep-phyloMI*). *Phyloscanner* [38], which analyzes within-host pathogen diversity from deep-sequencing reads, was used to infer within-host phylogenetic trees across the HIV genome, remove contaminant sequences, and identify regions of the genome with evidence of multiple infecting variants. Our model simultaneously estimates individual- and population-level risks of harboring a multiple infection from processed *phyloscanner* output after accounting for incomplete sequencing of the viral population within a sample and false-negative and false-positive rates of multiple variant identification. We validated model performance on simulated data and used it to identify multiple infections in RCCS participants over a period of declining incidence and rapidly shifting transmission dynamics [35,39].

## 2. Materials and methods

### 2.1. Ethics statement

All participants provided written informed consent for the study. Written assent and written parental consent were obtained for participants less than 18 years of age. The RCCS is administered by the Rakai Health Sciences Program (RHSP) and has received ethical approval from the Uganda Virus Research Institute's Research and Ethics Committee (GC/127/08/12/137), the Uganda National Council for Science and Technology (HS450), and the Johns Hopkins School of Medicine (IRB00217467).

### 2.2. Study design and participants

The RCCS conducts population-based surveys every 18–24 months in agrarian, semi-urban trading, and Lake Victoria fishing communities in southern Uganda. Data in this study were collected over six RCCS survey rounds conducted between January 2010 and November 2020. As survey rounds occurred over more than a year, we herein refer to them by the median interview date. Communities that participated in the RCCS were categorized based on their geographic setting and primary economic activity (inland communities: agrarian/trading, Lake Victoria communities: fishing). These communities differ considerably in their HIV burden (HIV prevalence of ~14% [agrarian], ~17% [trading], and ~42% [fishing]) [36]. At each survey round, households were censused and all residents aged 15–49 who were able to provide consent (assent for those under 18) were invited to participate in a survey. Survey participants were eligible to participate exactly once in each survey round ("participant-visits"). As part of the survey, participants completed a detailed structured sociodemographic, behavioral, and health questionnaire. Specifically, participants were asked to self-report their sex, age, residency status (e.g. recent migration into a community), circumcision status (among males), occupation, occupation of sex partners in the year prior to the survey, and number of lifetime sex partners. As HIV is more prevalent among female sex and bar/restaurant workers [40,41], we generated a composite variable indicating reported sex or bar/restaurant work among women and sex with a sex or bar/restaurant worker among men to determine if these individuals were at higher risk of being multiply infected.

To account for the fact that the number of lifetime sex partners increases over the lifespan, we calculated the mean number of lifetime sex partners within population strata ($s$) defined by HIV serostatus, sex, age category in five year bins, and community type (inland/fishing) ($\bar{P}_s$) to allow for standardization of the observed responses. Responses of no lifetime sex

partners were treated as missing data as HIV transmission in this setting is predominantly heterosexual [42] and we therefore expected these individuals to have had at least one sexual encounter in order to acquire HIV, although we cannot rule-out perinatal transmission with available data. When calculating $\bar{P}_s$ missing data was imputed to the mean value of a lognormal distribution fit to all numeric responses of $\geq 1$ lifetime sex partner within strata defined by HIV serostatus, sex, age category, and community type. Additionally, some RCCS participants provided categorical responses ("1–2" or "3+" lifetime sex partners). To calculate $\bar{P}_s$, we first imputed these values to a numeric response. Responses of "1–2" were imputed to the mean response among PLHIV reporting either one or two lifetime sex partners within strata. Similarly, responses of "3+" were imputed to the mean value of a lognormal distribution fit to all numeric responses of $\geq 3$ lifetime partners within strata as above.

In addition to completing the survey questionnaire, participants provided venous blood samples for HIV testing, viral load quantification, and viral deep sequencing. HIV serostatus was evaluated using a validated rapid test algorithm [43]. HIV viral load quantification was conducted using the Abbott real-time m2000 assay (Abbott Laboratories).

## 2.3. HIV deep sequencing and bioinformatic processing

HIV RNA deep-sequence data from plasma samples contributed by RCCS participants was generated through the Phylogenetics and Networks for Generalized HIV Epidemics in Africa consortium (PANGEA-HIV) [44–46]. The study sample included RCCS participants with HIV who were viremic ($\geq 1,000$ copies/mL) at one of their study visits between January 2010 and November 2020. To avoid biasing our inferences, for individuals that participated in multiple survey rounds we used only the data from the sample with the highest genome coverage or the highest viral load in the case of ties in our analyses of multiple infections. The study sample was further restricted to individuals in putative transmission networks and excluded individuals for who another phylogenetically close individual could not be identified over the entire study period [39]. All available sequence data for individuals in putative transmission networks was included in phylogenetic analyses.

Deep-sequencing was performed with two protocols (S1 Table), as previously described [39]. Briefly, for sequence data generated through the amplicon protocol, viral RNA was extracted from plasma samples on the QIAsymphony SP workstation with the QIAsymphony DSP Virus/Pathogen Kit. cDNA was generated through one-step reverse transcription PCR protocol using universal HIV-1 primers designed to generate four overlapping amplicons across the HIV-1 genome [34]. Deep-sequencing was conducted at the Wellcome Trust Sanger Institute core facility using the Illumina MiSeq and HiSeq platforms. To generate sequence data using the bait-capture protocol viral RNA was similarly extracted using the QIAsymphony DSP Virus/Pathogen Kit followed by library preparation according to the veSEQ-HIV protocol [33]. Library preparation was performed using the SMARTer Stranded Total RNA-Seq v2-PicoInputMammalian (Clontech, TakaRaBio) kit and double-stranded dual-indexed cDNA generated using in-house indexed primers. Libraries were pooled and cleaned with Agencourt AMPure XMP. Pooled libraries were hybridized to HIV-specific biotinylated 120-mer oligonucleotides (xGen Lockdown Probes, Integrated DNA Technologies) and isolated with streptavidin-conjugated beads. Captured libraries were PCR amplified prior to generation of 350-600 base pair (bp) paired-ends reads with the Illumina NovaSeq 6000 at the Oxford Genomic Centre.

Kraken v.0.10.5-beta [47] with a custom database of human, bacterial, archaeal, viral, and fungal genomes was used to isolate reads of viral and unknown origin which were trimmed of adaptors and low-quality bases using trimmomatic [48] v.0.36/0.39. Trimmed reads

were *de novo* assembled into contigs using SPAdes [49] and metaSPAdes [50] v.3.10. Shiver v.1.5.7 [51] was used to align reads to a reference sequence constructed for each sample using these contigs.

## 2.4. Inference of within-host deep-sequence phylogenetic trees

To improve the computational efficiency of our within-host deep-sequence phylogenetic analyses we first clustered participants with HIV into putative transmission networks as previously described (S1 File) [39,52], and then grouped putative networks into batches for deep-sequence phylogenetic analyses.

Deep-sequence data belonging to participants in each batch were further processed with *phyloscanner* [38] v.1.8.1 to infer within-host phylogenetic trees in 287 sliding windows of length 250 bp with a step size of 25 across the HIV genome as in [39]. As suggested in [38], this window-size was chosen to be long enough to capture sufficient within-host diversity to provide phylogenetic signal but no longer than the target read length and short enough to minimize within-window recombination. Windows spanning *env* gp120 were excluded as genetic diversity in the variable loop regions [53] led to poor sequence alignment and unreliable within-host phylogenetic trees. In addition to deep-sequence data from RCCS participants, we included as phylogenetic background 113 consensus sequences from representative subtypes and circulating forms and 200 near full-length consensus sequences from Kenya, Uganda, and Tanzania (Los Alamos National Laboratory HIV Sequence Database, http://www.hiv.lanl.gov, S2 File). Within *phyloscanner*, MAFFT v.7.475 [54] with iterative refinement and iterative re-alignment using consistency scores was used to align sequencing reads and IQ-TREE v.2.0.3 with the GTR+F+R(Free-Rate)6 substitution model was used for phylogenetic inference [55,56]. Phylogenetic branch lengths within *phyloscanner* were adjusted to account for varying substitution rates across the HIV genome as described in [57] (S3 File). Adjusted distances can be interpreted as average distances expected in the *pol* gene. The genomic coordinates of input sequence data were standardized to the coordinates of the HIV-1 HXB2 reference genome (GenBank: K03455.1).

For each participant, *phyloscanner* was used to estimate the number of genetically distinct phylogenetic lineages (subgraphs) in each genome window using a modified parsimony algorithm. In each window, for each participant, the given phylogenetic tree was pruned to include only tips from the given participant and the specified outgroup (here, the subtype H consensus sequence). Ancestral nodes in the pruned tree were assigned to one of two states: either that of the participant or an unsampled "unassigned" state (to which the outgroup and root of the phylogeny was assigned), representing the lineages that are evolutionarily ancestral to the lineages that initiated a given host's infection. To accurately assign nodes without relying on patterns of phylogenetic clustering with reference sequences, we employed a modified Sankoff minimum parsimony algorithm for ancestral state reconstruction as described in [38, 58] (in particular, see Supplementary Information 1.2 and Supplementary Fig 1 in [38]). This algorithm assigns a cost ($c(n, h)$) to a state change along a lineage ending at ancestral node $n$ that is proportional to the sum of the branch lengths descendant from that node that give rise to tips form host $h$ ($l(n, h)$). As tips from all other subjects with the exception of the outgroup were pruned from the tree prior to this procedure ("single-host tree"), this is equivalent to the sum of the total branch length of the subtree with node $n$ as its root. Specifically, this cost was calculated as:

$$c(n, h) = 1 + k \times l(n, h), \tag{1}$$

where $k$ is a tuneable constant that controls the penalty associated with fewer host $h$ subgraphs. Traditional parsimony is recovered when $k = 0$ which will always assign all tips in

a single-host tree to a single subgraph, regardless of the phylogenetic branch length captured within that subgraph. As $k \to \infty$, each tip belonging to host $h$ will be assigned to a unique subgraph. Here, we parameterized $k$ with the goal of distinguishing evolution that occurred within a given host from evolution that occurred prior to HIV acquisition, in the case of multiple infection. In the case of single infection, all tips in a single-host tree will be closely related (e.g. Fig 1A) and therefore we want the ancestral reconstruction that minimizes $c(n, h)$ to assign all tips to a single subgraph. In the case of a multiple infection the tips will be expected to fall into ($\geq$)2 clades with relatively small within-subgraph distances but large between-subgraph distances and we seek to parameterize $k$ such that the ancestral reconstruction minimizing $c(n, h)$ differentiates these clades into distinct subgraphs. We conservatively used a $k$ value of 15 such that $\frac{1}{k} = 0.067$, which is greater than the 99th percentile of the pairwise genetic distances between epidemiologically confirmed HIV transmission pairs [57] and comparable to within-subtype HIV genetic diversity within Rakai [7].

Quality filtering of inferred within-participant phylogenetic trees was performed with *phyloscanner*. Specifically, within each window, subgraphs with less than three reads or less than 1% of reads from a particular participant were marked as putative contaminants and removed from the analysis. To mask regions with insufficient data for reliable phylogenetic inference any window with less than 30 reads from a given participant after aforementioned filter was also removed from the analysis. After filtering we identified the subgraphs with data from the deep-sequenced reads from each sequenced sample for a given participant.

## 2.5. Bayesian model to identify multiple infections

We developed a Bayesian statistical model to identify samples harboring multiple infections and estimate the prevalence of multiple infections in a set of deep-sequencing reads that were processed with *phyloscanner*. We refer to this model as the the deep-phylo multiple infection model (*deep-phyloMI*). We first summarized the *phyloscanner* output for each sample and each genomic window in terms of two binary variables, $N^{obs}_{i=1...n,w}$ (presence/absence of sequencing reads from sample $i$ in window $w$ following *phyloscanner* contamination filtering) and $M^{obs}_{i=1...n,w}$ (presence/absence of multiple subgraphs for sample $i$ in window $w$) where $n$ is the number of sequenced samples. To simplify notation below, when $N^{obs}_{i,w} = 0$ we set $M^{obs}_{i,w} = 0$.

We further summarized the data for sample $i$ into two quantities, $N^{obs}_i = \sum_{w=1}^{n^{max}} N^{obs}_{i,w}$ and $M^{obs}_i = \sum_{w=1}^{n^{max}} M^{obs}_{i,w}$ where $n^{max}$ is the number of genome windows.

### 2.5.1. Base model accounting for partial sequencing success of infecting variants

We first developed a base model that accounts for partial sequencing success across the HIV genome in giving rise to the observed $N^{obs}_{i=1...n,w}$ and $M^{obs}_{i=1...n,w}$. Working from first principles, we first derived a likelihood model for observing the pair of counts $(N^{obs}_i, M^{obs}_i)$ for the unobserved groups of samples with true multiple infection ($M_i = 1$) and single infection ($M_i = 0$), and subsequently marginalise out the unknown true multiple infection status (either $M_i = 0$ or $M_i = 1$). Among samples from multiply infected individuals ($M_i = 1$), we assumed that the probability of sequencing each of the infecting variants in window $w$ was given by $\theta_i$ for each sample $i$. The probability of sequencing at least one variant in each window is therefore $1 - (1 - \theta_i)^2$ and the probability of sequencing both variants given at least one was sequenced is therefore $\frac{\theta_i}{2 - \theta_i}$. Assuming sequencing success was independently and identically distributed for each sample, we obtained

$$\left(N^{obs}_i | \theta_i, M_i = 1\right) \sim \text{Binomial}^{1+}\left(n^{max}, 1 - (1 - \theta_i)^2\right) \quad (2a)$$

$$\left(M_i^{\mathrm{obs}}|N_i^{obs},\theta_i,M_i=1\right) \sim \mathrm{Binomial}\left(N_i^{obs},\frac{\theta_i}{2-\theta_i}\right), \tag{2b}$$

where $\mathrm{Binomial}^{1+}$ represents the 0-truncated Binomial distribution as we only consider data from individuals with *phyloscanner* output in at least one genomic window, and $n^{\mathrm{max}}$ is the total number of genomic windows. This model implicitly accounts for the presence of windows in which only a single variant was present in the *phyloscanner* output due to incomplete sequencing success. For samples from individuals infected with only a single variant ($M_i = 0$), we obtained analogously

$$\left(N_i^{obs}|\theta_i,M_i=0\right) \sim \mathrm{Binomial}^{1+}\left(n^{\mathrm{max}},\theta_i\right) \tag{3a}$$

$$\left(M_i^{\mathrm{obs}}|N_i^{obs},\theta_i,M_i=0\right) \sim \mathrm{Binomial}\left(N_i^{obs},0\right). \tag{3b}$$

Taken together, the joint likelihood of observing the count pair $(M_i^{\mathrm{obs}}, N_i^{obs})$ conditional on latent multiple infection status $M_i$ is given by

$$P(N_i^{obs},M_i^{\mathrm{obs}}|\theta_i,M_i) = P(N_i^{obs}|\theta_i,M_i)P(M_i^{\mathrm{obs}}|N_i^{obs},\theta_i,M_i). \tag{4}$$

Thus, aggregating over the two unknown possible multiple infection states $M_i \in \{0,1\}$ for each sample in a finite mixture model framework, we have

$$\begin{aligned}P(N_i^{obs},M_i^{\mathrm{obs}}|\theta_i) = \\ \sum_{m=0,1} P(N_i^{obs}|\theta_i,M_i=m)P(M_i^{\mathrm{obs}}|N_i^{obs},\theta_i,M_i=m)P(M_i=m).\end{aligned} \tag{5}$$

One of our primary inferential targets was the individual-level probability of harboring multiple infection not conditional on observed $N^obs_i$ and $M_i^{\mathrm{obs}}$, which we denoted with $\delta_i = P(M_i=1)$. Making this target explicit in the joint likelihood, we have

$$P(N_i^{obs},M_i^{\mathrm{obs}}|\theta_i,\delta_i) = \tag{6a}$$

$$\delta_i \times P(N_i^{obs}|\theta_i,M_i=1)P(M_i^{\mathrm{obs}}|N_i^{obs},\theta_i,M_i=1) \tag{6b}$$

$$+ (1-\delta_i) \times P(N_i^{obs}|\theta_i,M_i=0)P(M_i^{\mathrm{obs}}|N_i^{obs},\theta_i,M_i=0), \tag{6c}$$

and so the log posterior distribution of the parameters $\theta = (\theta_1,\dots,\theta_n)$, $\delta = (\delta_1,\dots,\delta_n)$ for all the data $\mathbf{x} = \left((N_1^{obs},M_1^{\mathrm{obs}}),\dots,(N_n^{obs},M_n^{\mathrm{obs}})\right)$ under our model is

$$\log f(\theta,\delta|\mathbf{x}) \propto \sum_{i=1}^{n}\left(\log P(N_i^{obs},M_i^{\mathrm{obs}}|\theta_i,\delta_i) + \log f(\theta_i,\delta_i)\right), \tag{7}$$

where we use $f$ to denote posterior and prior densities.

**2.5.2. Base model prior densities** In the base model, prior to observing data, we modelled the individual-level probability of multiple infection as identical for all $i$ with the prior density,

$$\mathrm{logit}\left(\delta_i\right) = \delta_0 \sim \mathrm{Normal}(0,3.16^2), \tag{8}$$

with diffuse variance [59]. Given the known log-linear dependency of sequencing success on log viral load [33], known differences in sequencing success rates by sampling protocol [39],

and other factors, we specified the prior on the individual-level sequencing probability $\theta_i$ through a logistic mixed effects model. Specifically, we modeled $\text{logit}(\theta_i)$ with

$$\text{logit}(\theta_i) = \alpha_0 + \sigma_{\text{stz}}\alpha_1 X_i^{\text{amplicon}} + \sigma_{\text{stz}}\alpha_2 X_i^{\text{bait}} + \tag{9a}$$

$$\alpha_3 V_i + \alpha_4 V_i^a X_i^{\text{amplicon}} + \alpha_5 V_i^b X_i^{\text{bait}} + \alpha_i \tag{9b}$$

$$\alpha_0 \sim \text{Normal}(0, 2^2) \tag{9c}$$

$$\alpha_3 \sim \text{Normal}(0, 2^2) \tag{9d}$$

$$(\alpha_1, \alpha_2) \sim \sigma_{\text{stz}} \times \text{stz-MVN}(0, 1) \tag{9e}$$

$$(\alpha_4, \alpha_5) \sim \sigma_{\text{stz}} \times \text{stz-MVN}(0, 1) \tag{9f}$$

$$\alpha_i \sim \text{Normal}(0, \sigma_{\text{ind}}^2) \tag{9g}$$

$$\sigma_{\text{ind}} \sim \text{Half-Cauchy}(0, 1), \tag{9h}$$

where $X_i^{\text{amplicon}}$ and $X_i^{\text{bait}}$ are indicator variables for whether sample $i$ was sequenced using the amplicon or bait capture approach respectively and $V_i$, $V_i^a$, and $V_i^b$ are the sample $\log_{10}$ copies/mL values standardized to have mean zero and standard deviation 1 among all samples and among only the amplicon ($V_i^a$) and bait capture ($V_i^b$) samples, respectively. To maintain identifiability we constrain $\alpha_1 + \alpha_2 = \alpha_4 + \alpha_5 = 0$ by specifying their joint prior distributions with a zero-mean multivariate normal with a particular variance-covariance matrix described in [59], such that all marginal distributions are standard normal, e.g. $\alpha_1 \sim \text{Normal}(0, 1)$ and $\alpha_2 \sim \text{Normal}(0, 1)$, which we represent with the notation stz-MVN. To maintain marginal priors with standard deviation $\sigma_{\text{stz}} = 2$, we adopt a non-centered parameterisation and post-multiply the sum-to-zero random variables with $\sigma_{\text{stz}}$. Finally, $\alpha_i$ denotes an individual-level random effect.

**2.5.3. Modelling false-negative and false-positive phylogenetic observations** We extended the base model to account for possible false-negative and false-positive phylogenetic observations, accounting for incomplete removal of false-positive observations through *phyloscanner*, and/or incomplete phylogenetic identification of multiple infections due to insufficient phylogenetic background. First, among samples from individuals in which $M_i = 1$ we accounted for the scenario in which both variants are successfully sequenced in a given window but were identified as a single phylogenetic clade by *phyloscanner*, i.e. false-negative observations, by modifying our data-generating model to

$$\left(M_i^{\text{obs}}|N_i^{obs}, \theta_i, M_i = 1\right) \sim \text{Binomial}\left(N_i^{obs}, \frac{\theta_i}{2 - \theta_i}(1 - \lambda)\right), \tag{10}$$

where $\lambda$ represents the false-negative rate. We analogously accounted for the scenario in which only a single variant was sequenced but *phyloscanner* spuriously assigned multiple subgraphs in a given window, i.e. false-positive observations, through a false-positive rate $\epsilon$ in the model. We modeled false-positives among samples lacking multiple infection and among windows in multiply infected samples in which only a single variant was sequenced, which occurs with probability $2\frac{1-\theta_i}{2-\theta_i}$ when $N_{i,w} = 1$, but was spuriously assigned to two subgraphs. Note that because we did not differentiate between windows with exactly 2 and >2 subgraphs, we do not consider the scenario where both variants are sequenced in a true multiple infection and the two sequenced variants are spuriously assigned to 3 or 4 subgraphs. Our data generating model was updated to account for false-positives and

false-negatives as:

$$\left(M_i^{\text{obs}}|N_i^{obs}, \theta_i, M_i = 1\right) \sim \text{Binomial}\left(N_i^{obs}, \frac{\theta_i}{2 - \theta_i}(1 - \lambda) + 2\epsilon\frac{1 - \theta_i}{2 - \theta_i}\right) \tag{11a}$$

$$\left(M_i^{\text{obs}}|N_i^{obs}, \theta_i, M_i = 0\right) \sim \text{Binomial}\left(N_i^{obs}, \epsilon\right), \tag{11b}$$

with additional prior densities

$$\text{logit}(\lambda) \sim \text{Normal}(0, 1)[, 2.2] \tag{12a}$$

$$\text{logit}(\epsilon) \sim \text{Normal}(0, 1), \tag{12b}$$

where $[, 2.2]$ represents that $\text{logit}(\lambda)$ was constrained to be $<2.2$ and all other components of the model remaining as above.

### 2.5.4. Estimating risk factors of multiple infection

We further extended the model described above to model the probability of multiple infection as dependent on potential clinical, behavioral, and/or epidemiological risk factors through a logistic regression approach. Specifically, we modeled the logit of the individual-level multiple infection prior probabilities as a linear predictor of fixed effects,

$$\text{logit}\left(\delta_i\right) = X_i^{\text{risk}}\beta = \delta_0 + \sum_{j=1}^{n_c} X_i^j\beta_j \tag{13a}$$

$$\beta_j \sim \text{Normal}(0, 1), \ \text{if} \ k_j = 1 \tag{13b}$$

$$\beta_j \sim \text{stz-MVN}(0, 1), \ \text{if} \ k_j > 1, \tag{13c}$$

where $X_i^j$ are $1 \times k_j$ dimensional row vectors for each of $n_c$ putative multiple infection predictive covariates and $\beta_j$ are $k_j \times 1$ dimensional column vectors of fixed effect coefficients. For all categorical $j$ in $n_c$ with $k_j$ levels, we model the corresponding $k_j$ fixed effects with the sum-to-zero joint multivariate normal prior defined above to maintain identifiability.

We also considered a fixed effects model with Horseshoe-type shrinkage priors [60,61] on the effect sizes to handle correlated individual-level covariates. To maintain desirable sum-to-zero properties, we define a global non-negative shrinkage parameter $\tau \in [0, \infty)$, and for each categorical $j$ with $k_j$ levels $k_j$ non-negative local shrinkage parameters $\xi_j \in [0, \infty)^{k_j}$, and the diagonal matrix $D_j = \text{diag}(\xi_j)$. We then specify $k_j$ sum-to-zero shrinkage effects $\beta_j$ through a joint zero-mean multivariate normal distribution with variance covariance matrix $\frac{k_j}{k_j-1}[D_j - D_j 1(1^T D_j 1)^{-1} 1^T D_j]$, such that $0 = \sum_{l=1}^{k_j} \beta_{j,l}$ and the induced marginal distributions of each $\beta_{j,l}$ are $\text{Normal}(0, \xi_{j,l}^2)$, which we refer to stz-MVN$(0, \xi_j^2)$. We incorporated the global shrinkage parameter in non-centered parameterisation through post-multiplication as in Eq 9. Therefore, we have:

$$\text{logit}\left(\delta_i\right) = X_i^{risk}\beta = \delta_0 + \sum_{j=1}^{n_c} X_i^j\beta_j \tag{14a}$$

$$\beta_j|\xi_j, \tau \sim \tau \times \text{Normal}(0, \xi_j^2), \ \text{if} \ k_j = 1 \tag{14b}$$

$$\beta_j|\xi_j, \tau \sim \tau \times \text{stz-MVN}(0, \xi_j^2), \ \text{if} \ k_j > 1 \tag{14c}$$

$$\xi_j \sim \text{Half-}t_2(0, 1) \tag{14d}$$

$$\tau \sim \text{Half-Cauchy}(0, 1), \tag{14e}$$

where we modelled the $\xi_j$ with t-distributions with 2 degrees of freedom instead of Cauchy distributions to ease numerical sampling.

As above, the number of lifetime sex partners included missing and ambiguous responses (e.g. "3+"), and these values were estimated as additional random variables in the Bayesian inference, assuming they were missing at random within sex, age, and community type, using lognormal prior distributions specific to these strata defined by the non-missing responses as above. Imputed values for missing responses were limited to the range [1,60] and responses of "3+" were limited to the range [3,60].

**2.5.5. Parameter estimation** We estimated joint posterior distributions numerically using Hamiltonian Monte Carlo [62] with the No-U-Turn Sampler [63] implemented in Stan [59] and accessed through cmdStanR v.2.36.0 [64] in R. For all analyses, four independent chains with 2,000 iterations of warm up and 2,000 iterations of sampling were run. A target acceptance rate of 0.8 was used for all analyses with the exception of those that employed shrinkage priors where a target acceptance rate of 0.95 was used to avoid divergent transitions. Convergence was assessed using the $\hat{R}$ statistic, bulk and tail effective sample sizes (ESS) for each parameter [65], and visual inspection of trace and pairs plots.

**2.5.6. Generated quantities** Based on the estimated parameter distributions of the models described above, we generated a number of quantities to aid in interpretation of our results.

*2.5.6.1. Posterior probabilities of individual-level multiple infection.*

We computed the posterior probabilities of individual-level multiple infection directly from Monte Carlo samples of the joint posterior density via

$$P(M_i = 1|N_i^{obs}, M_i^{\mathrm{obs}}, n^{\mathrm{max}}) =$$
$$\int P(M_i = 1|N_i^{obs}, M_i^{\mathrm{obs}}, n^{\mathrm{max}}, \theta_i, \delta_i, \lambda, \epsilon) P(\theta_i, \delta_i, \lambda|N_i^{obs}, M_i^{\mathrm{obs}}) d(\theta_i, \delta_i, \lambda), \tag{15}$$

by taking for each individual $i$ all Monte Carlo samples of the posterior density of $(\theta_i, \delta_i, \lambda)$, evaluating $P(M_i = 1|N_i^{obs}, M_i^{\mathrm{obs}}, n^{\mathrm{max}}, \theta_i, \delta_i, \lambda, \epsilon)$ according to:

$$P(M_i = 1|N_i^{obs}, M_i^{\mathrm{obs}}, n^{\mathrm{max}}, \theta_i, \delta_i, \lambda, \epsilon) = \frac{\delta_i \times P(N_i^{obs}, M_i^{\mathrm{obs}}|n^{\mathrm{max}}, \theta_i, \lambda, \epsilon, M_i = 1)}{P(N_i^{obs}, M_i^{\mathrm{obs}}, |n^{\mathrm{max}}, \theta_i, \delta_i, \lambda, \epsilon)}, \tag{16}$$

and calculating the expectation across these.

*2.5.6.2. Prevalence of multiple infection in the study sample.*

Following from prior work on Bayesian latent class models with covariates [66–71], under the base model the posterior estimate of the prevalence of multiple infection in the study sample is given by:

$$\bar{\delta} = \text{inverse-logit}\,(\delta_0) = \frac{\exp(\delta_0)}{1 + \exp(\delta_0)}, \tag{17}$$

where $\delta_0$ is from the joint posterior density of the model defined by Eqs 8, 9, 11, and 12. In the presence of modeled risk factors, the prevalence of multiple infections in the study sample will vary based on sub-groups $s$ defined by $X^{risk}$. In the case where $X^{\mathrm{risk}}$ contains only the covariates used to define $s$:

$$\bar{\delta}_s = \text{inverse-logit}\,\left(\delta_0 + X_s^{\mathrm{risk}}\beta\right). \tag{18}$$

Finally, we estimated the prevalence in a target population (e.g. the entire sample of sequenced viremic RCCS participants) through post-stratification:

$$\bar{\delta} = \frac{\sum_{s=1}^{S} Q_s \bar{\delta}_s}{\sum_{s=1}^{S} Q_s},$$ (19)

where $Q_s$ are the number of sampled individuals in each of the $S$ sub-populations $s$ and $\bar{\delta}_s$ are the sub-group specific prevalence estimates from Eq 18.

### 2.5.6.3. Prevalence and risk ratios of harboring multiple infection associated with epidemiological covariates.

We calculated a posterior estimate for the prevalence risk ratio (PRR) of multiple infections in epidemiological strata $s^*$ as compared to strata $s$ as

$$\mathrm{PRR}_{s*,s} = \frac{\bar{\delta}_{s*}}{\bar{\delta}_s}.$$ (20)

In the case where $X^{\mathrm{risk}}$ contained additional covariates beyond those used to define $s^*$ from $s$ we estimated a multivariate risk ratio (RR) associate with the covariate(s) that distinguish $s^*$ from $s$ by calculating the ratio of the estimated risk of multiple infection for person $i$ as if they belonged to strata $s^*$ divided by the risk of multiple infection of the same person $i$ as if they belonged to strata $s$, while holding all other covariates at their observed values (based on the design matrices $X^{\mathrm{risk}}_{i|i \in s*}$ and $X^{\mathrm{risk}}_{i|i \in s}$, respectively):

$$\mathrm{RR}_{s*,s} = \frac{1}{n} \sum_{i=1}^{n} \frac{\mathrm{inverse\text{-}logit}\left(\delta_0 + X^{\mathrm{risk}}_{i|i \in s*}\beta\right)}{\mathrm{inverse\text{-}logit}\left(\delta_0 + X^{\mathrm{risk}}_{i|i \in s}\beta\right)}.$$ (21)

### 2.5.6.4. Post-stratification adjustments.

Finally, because sequence data was not available for all viremic participants with HIV in our study population, we employed post-stratification based on prevalence estimates in epidemiological sub-groups $s$ to estimate the prevalence of multiple infections in the population under study (viremic study participants) [72]. Specifically, we calculated

$$\bar{\delta}^* = \frac{\sum_{s=1}^{j} W_s \bar{\delta}_s}{\sum_{s=1}^{j} W_s},$$ (22)

where $W_s$ is the estimated population size or estimated relative population size of sub-group $s$. The population prevalence ratio between two non-overlapping composite sub-groups can therefore be calculated as in Eq 20. We performed post-stratification based on the total number of participant-visits from viremic PLHIV stratified by age ((14, 24], (24, 34], and (34, 49] years), sex, and community type. Because viral load measurements were not routinely conducted for all PLHIV in the 2010 and 2012 survey rounds we calculated population-sizes using only participant-visits in the 2014-2019 survey rounds.

## 2.6. Simulation study

We used simulations to validate our inference model. For all simulations, we simulated data for $n^{\max} = 29$ genome windows in $n = 2,000$ samples which were assigned a normalized $\log_{10}$ viral load ($V_i$) with random draws from a $N(0,1)$ distribution. For all samples, $\alpha_i$ was drawn from a $N(0,1)$ distribution and $\theta_i$ calculated as $\alpha_0 + \alpha_1 V_i + \alpha_i$ with $\alpha_0 = 2$ and $\alpha_1 = 2$. Under these parameters, we generated three simulated data sets as described below.

### 2.6.1. Base simulation

$$M_i = \left[1_{\times 100}\right] \oplus \left[0_{\times 1900}\right] \tag{23a}$$

$$\left(N_i^{obs} | M_i = 0\right) \sim \text{Binomial}^{1+}(29, \phi_i) \tag{23b}$$

$$\left(N_i^{obs} | M_i = 1\right) \sim \text{Binomial}^{1+}\left(29, \left(1 - (1 - \phi_i)^2\right)\right) \tag{23c}$$

$$\left(M_i^{\text{obs}} | M_i = 0\right) \sim \text{Binom}\left(N_i^{obs}, 0\right) \tag{23d}$$

$$\left(M_i^{\text{obs}} | M_i = 1\right) \sim \text{Binomial}\left(N_i^{obs}, \frac{\phi_i}{2 - \phi_i}\right), \tag{23e}$$

where $\left[x_{\times n}\right]$ represents a vector of $x$ repeated $n$ times and $\oplus$ represents concatenation of two vectors.

### 2.6.2. Full simulation

$$M_i = \left[1_{\times 100}\right] \oplus \left[0_{\times 2000}\right] \tag{24a}$$

$$\left(N_i^{obs} | M_i = 0\right) \sim \text{Binomial}^{1+}(29, \phi_i) \tag{24b}$$

$$\left(N_i^{obs} | M_i = 1\right) \sim \text{Binomial}^{1+}\left(29, \left(1 - (1 - \phi_i)^2\right)\right) \tag{24c}$$

$$\left(M_i^{\text{obs}} | M_i = 0\right) \sim \text{Binom}\left(N_i^{obs}, \epsilon\right) \tag{24d}$$

$$\left(M_i^{\text{obs}} | M_i = 1\right) \sim \text{Binom}\left(N_i^{obs}, \frac{\phi_i}{2 - \phi_i}(1 - \lambda) + 2\epsilon \frac{1 - \phi_i}{2 - \phi_i}\right) \tag{24e}$$

$$\lambda = 0.30 \tag{24f}$$

$$\epsilon = 0.01. \tag{24g}$$

Additional simulations from this simulation model were generated with all other parameters held constant except (A): $\sum_i^n M_i = 0, 300, 600$, (B): $\lambda = 0.10, 0.20, 0.40$, and (C): $\epsilon = 0, 0.005, 0.05$.

### 2.6.3. Extended simulation

$$M_i = \left[1_{\times 150}\right] \oplus \left[0_{\times 1850}\right] \tag{25a}$$

$$X_{,1}^{\text{risk}} = \left[1_{\times 100}\right] \oplus \left[0_{\times 50}\right] \oplus \left[1_{\times 900}\right] \oplus \left[0_{\times 950}\right] \tag{25b}$$

$$X_{i,2-5}^{\text{risk}} \sim \text{shuffle}\left(\left[1_{\times 1000}\right] \oplus \left[0_{\times 1000}\right]\right) \tag{25c}$$

$$\left(N_i^{obs} | M_i = 0\right) \sim \text{Binomial}^{1+}(29, \phi_i) \tag{25d}$$

$$\left(N_i^{obs} | M_i = 1\right) \sim \text{Binomial}^{1+}\left(29, \left(1 - (1 - \phi_i)^2\right)\right) \tag{25e}$$

$$\left(M_i^{\text{obs}} | M_i = 0\right) \sim \text{Binomial}\left(N_i^{obs}, \epsilon\right) \tag{25f}$$

$$\left(M_i^{\text{obs}} | M_i = 1\right) \sim \text{Binomial}\left(N_i^{obs}, \frac{\phi_i}{2 - \phi_i}(1 - \lambda) + 2\epsilon \frac{1 - \phi_i}{2 - \phi_i}\right), \tag{25g}$$

where $X^{\text{risk}}_{i,j}$ represents the entry in the *ith* row and *jth* column of the design matrix $X^{\text{risk}}$ and shuffle($v$) denotes shuffling the elements of $v$.

## 2.7. Data analysis and visualization

All data analysis was conducted in R v.4.4.1 [73] using the tidyverse [74] with dplyr v.1.1.4 [75], tibble v.3.2.1 [76], and tidyr v.1.3.1 [77]. Haven v.2.5.4 [78] was used to parse a subset of input data files. Visualization of data and results was done using ggplot2 v.3.5.1 [79] with bayesplot v.1.11.1 [80,81], cowplot v.1.1.3 [82], and patchwork v.1.2.0. [83]. Phylogenetic trees were manipulated and visualized using ape v.5.8 [84], ggtree v.3.12.0 [85–89], phytools v.2.1.-1 [90], and tidytree v.0.4.6 [85]. Highest posterior density intervals were calculated with HDInterval v.0.2.4 [91] and convergence statistics were assessed with posterior v.1.6.0 [92]. Preliminary analyses and model fitting was performed using fitdistrplus v.1.1-11 [93].

## 3. Results

### 3.1. Phylogenetic signatures of multiple infection in population-based pathogen surveillance

Between 2010 and 2020, 50,967 participants contributed to the RCCS in 109,608 visits over six survey rounds. Overall, 8,841 participants were HIV seropositive and 3,586 were viremic (plasma viral load ≥ 1,000 copies/mL) at one of their visits (S2 and S3 Tables). Of these, 2 ,029 individuals were sampled between January 2010 and November 2020, had HIV RNA deep-sequence data available at minimum quality criteria for deep-sequence phylogenetic analysis, and were identified as a member of a putative transmission network (Tables 1 and S4 and S1 File.). Availability of sequence data among viremic participants was generally higher among men, from residents of fishing communities, and from participants aged 25-34 years.

We next inferred within-host phylogenies from deep-sequencing reads in twenty-nine 250 bp non-overlapping genomic windows using *phyloscanner* (S4 File.), which captured evolutionary relationships of HIV variants within individual participants. Sequencing coverage varied significantly between samples (median [interquartile range (IQR)]: 5000 [4250] bp, S1A Fig) but was generally higher among bait capture sequenced samples and samples with higher viral load. Across the genome, sequencing success was highest in *gag* (Figs 1F and S1B), likely due to differential amplification efficiency of the primers used in the amplicon sequencing approach [94].

To characterise phylogenetic signatures of multiple infection, we used *phyloscanner* to identify distinct co-circulating variants among participants with viremic HIV (Materials and methods and Fig 1A and 1B). We tabulated the number ($M^{\text{obs}}_i$) of genome windows in which distinct phylogenetic lineages (phylogenetic subgraphs) were observed. The median genetic distance between the most recent common ancestors of subgraphs in genome windows with multiple subgraphs was 0.19 [IQR: 0.17] substitutions/site (Figs 1C and S2), which is consistent with contemporary circulating genetic diversity within Rakai [7,57]. Empirically, 181 (8.92%) samples had multiple subgraphs in at least one of the 29 non-overlapping windows (Fig 1D). Among these, the proportion of sequenced windows in which multiple subgraphs were observed varied considerably, but was generally relatively rare (median [IQR] 11.11 [19.14]% of sequenced windows for each sample, 2 [3] windows total). We observed a clear dependence of the ability to identify multiple subgraphs on sequencing success as quantified by genome coverage in the *phyloscanner* output. Of those samples with sequence data in all genome windows, 12.26% (52/424) had at least one window with multiple subgraphs compared to 8.04% (129 / 1605) among the remaining samples. Multiple subgraph

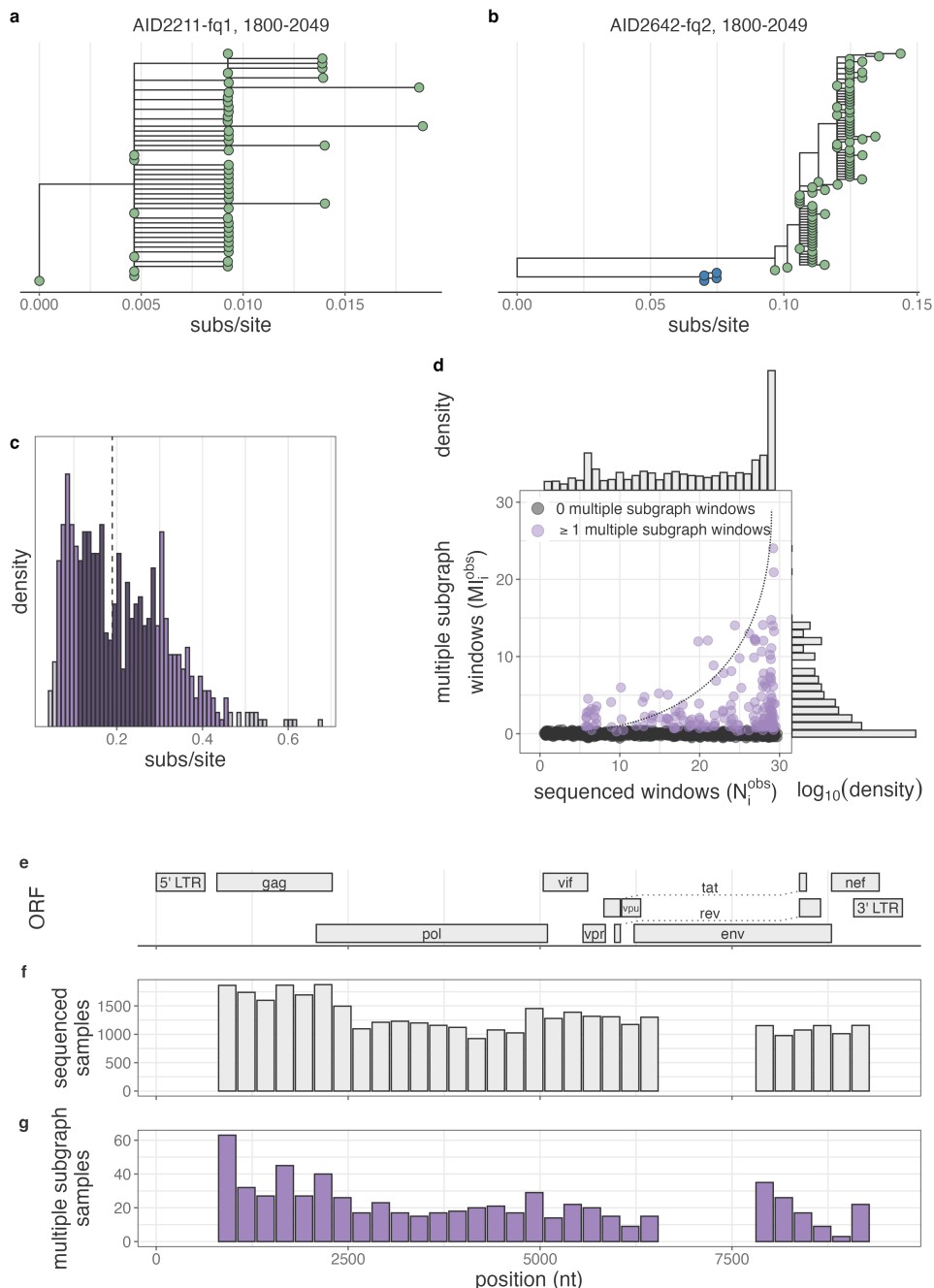

**Fig 1. Empiric phylogenetic multiple infection signatures from 2,029 samples from people with viremic HIV in the Rakai Community Cohort Study, 2010-2020.** (A) Representative within-host phylogenetic tree lacking evidence of multiple phylogenetic subgraphs. (B) Representative within-host phylogenetic tree with two subgraphs as indicated by the green and blue shading of the tips. (C) Distribution of branch length distance between the MRCAs of the two subgraphs with the most sequencing reads in all genome windows windows with $\geq 2$ subgraphs from all samples. Bins are shaded according to the 95th and 50th percentile. Vertical dotted line indicates median value. Binwidth is calculated such that there are approximately 50 bins across the range of observed values. (D) Per-sample number of non-overlapping genome windows with sequence data versus the number of non-overlapping genome windows with multiple subgraphs. Samples with at least one window with multiple subgraphs are shown in purple. Points have been jittered along both the X and Y axes for visual clarity. Dotted line shows modeled prediction in the absence of false-positive or false-negative multiple subgraph windows. Marginal densities are shown at right and above the scatter-plot. (E) Schematic of the HIV genome based on the coordinates from HXB2 (Genbank: K03455.1). (F) Number of samples with sequence data in each of the 29 non-overlapping genome windows. (G) Number of samples with evidence of multiple subgraphs in each of the 29 non-overlapping genome-windows.

**Table 1. Characteristics of the study sample obtained from population-level HIV deep-sequence surveillance in the Rakai Community Cohort Study, 2010-2020, stratified by availability of deep-sequence data.**

| | Participants living with HIV | | | |
|---|---|---|---|---|
| | | | Processed w/ PHSC | |
| | All | Viremic | n (%) | % of viremic (95% CI) |
| Overall | 8,841 | 3,586 | 2,029 | |
| Survey Round | | | | |
| 2010 | 1,812 (21.72%) | 35 (1.11%) | 16 (0.79%) | 45.71% (30.46%–61.82%) |
| 2012 | 2,202 (26.4%) | 992 (31.58%) | 749 (36.91%) | 75.5% (72.73%–78.08%) |
| 2014 | 1,170 (14.03%) | 653 (20.79%) | 346 (17.05%) | 52.99% (49.15%–56.79%) |
| 2015 | 1,292 (15.49%) | 727 (23.15%) | 523 (25.78%) | 71.94% (68.56%–75.09%) |
| 2017 | 1,080 (12.95%) | 511 (16.27%) | 328 (16.17%) | 64.19% (59.94%–68.23%) |
| 2019 | 786 (9.42%) | 223 (7.1%) | 67 (3.3%) | 30.04% (24.4%–36.37%) |
| Sex | | | | |
| Female | 5,315 (63.71%) | 1,685 (53.65%) | 1,032 (50.86%) | 61.25% (58.9%–63.54%) |
| Male | 3,027 (36.29%) | 1,456 (46.35%) | 997 (49.14%) | 68.48% (66.04%–70.81%) |
| Commiunity type | | | | |
| Inland | 4,974 (59.63%) | 1,212 (38.59%) | 742 (36.57%) | 61.22% (58.45%–63.92%) |
| Fishing | 3,368(40.37%) | 1,929(61.41%) | 1,287 (63.43%) | 66.72% (64.58%–68.79%) |
| Age | | | | |
| (14, 24] | 1,472 (17.65%) | 678 (21.59%) | 431 (21.24%) | 63.57% (59.88%–67.11%) |
| (24, 34] | 1,472 (46.45%) | 678 (50.21%) | 1,052 (51.85%) | 66.71% (64.34%–68.99%) |
| (34, 49] | 2,995 (35.9%) | 886 (28.21%) | 546 (26.91%) | 61.63% (58.38%–64.77%) |
| Viral load (log$_{10}$ copies/mL) | | | | |
| (3, 3.5] | | 466 (14.84%) | 275 (13.55%) | 59.01% (54.49%–63.39%) |
| (3.5, 4] | | 803 (25.57%) | 522 (25.73%) | 65.01% (61.64%–68.23%) |
| (4, 4.5] | | 806 (25.66%) | 521 (25.68%) | 64.64% (61.28%–67.86%) |
| (4.5, 5] | | 661 (21.04%) | 447 (22.03%) | 67.62% (63.96%–71.08%) |
| (5, ∞] | | 405 (12.89%) | 264 (13.01%) | 65.19% (60.42%–69.66%) |

For each participant, includes data from the participant-visit processed with PHSC if applicable or the participant-visit with the highest viral load, using the first visit in the case of ties or for people not living with HIV. Viremic participants excludes individuals living with HIV with suppressed viral load or missing viral load data. Viral load testing was not routinely conducted in earlier study rounds and was available for 37.32% of participant-visits contributed by people living with HIV in the 2010 and 2012 rounds. In recent rounds, viral load testing is routinely conducted and is available for 99.67% of participant-visits contributed by people living with HIV in the 2014-2019 surveys. Percentages represent the row percentages within each category. Binomial confidence intervals were calculated using the Agresti–Coull method. PHSC = phyloscanner.

windows were more common in the genome windows corresponding to *gag*, *env*, and *nef*, likely reflecting circulating genetic diversity in these regions with higher substitution rates [95]. Previous studies of HIV multiple infection in this setting have used amplicon-based deep-sequencing of two regions in p24 (1427–1816) and gp41 (7941–8264) regions [2,29,30]. Of 1,742 sequenced participants with data in windows spanning these regions (S4 File.), 75 (4.31%) had multiple subgraphs in one of the regions.

## 3.2. Bayesian model to identify multiple infections from pathogen deep-sequence data

The observed dependence between phylogenetically identified samples with multiple infection and successful genome sequencing implies it is difficult to deduce the underlying prevalence of multiple infections from the empirical data without a statistical model that accounts for partial sequencing success, false-positive multiple subgraphs, and false-negative unique subgraphs within hosts. Specifically, because identification of multiple infection requires successful sequencing of both variants and genetic divergence between those variants, there is

inherently more uncertainty in multiple infection status when sequencing success is poor or when infecting variants are genetically related in the sequenced region of the genome. Further, contamination or sequencing errors may give rise to spurious within-host genetic diversity and thereby inflate the estimated prevalence of multiple infection.

Therefore, we constructed a Bayesian model accounting for partial sequencing success to estimate the probabilities that each individual harbors a multiple infection, prevalence of multiple infection among deep-sequenced viremic participants, and risk factors for multiple infection (Materials and methods). We first verified that we were able to accurately estimate model parameters on simulated test data in the presence of incomplete sequencing success (S5 Table.). Next, we investigated the impact of false-positive and false-negative observations, as empiric analyses of RCCS deep-sequence data indicated that false-negative rates were likely substantial in that among samples with $M_i^{\text{obs}} > 0$, the observed $M_i^{\text{obs}}$ values for a given number of sequenced windows ($N_i^{obs}$) was less than expected based on our model (S1 File. and Fig 1D). We found that failing to account for these errors led to an overestimation of the prevalence of multiple infections on simulated data (Fig 2A and S6 Table.). This prompted us to explicitly include false-positive and false-negative detection rates in our model as free parameters. With this, we found that model parameters could be accurately estimated on simulated data (Fig 2B–2H and S7 Table.). Model performance was robust across simulations covering a range of reasonable values of the prevalence of multiple infections as well as false-positive and false-negative rates of multiple subgraph observation(S3, S4, and S5 Figs).

To identify risk factors for multiple infection among people living with HIV, we formulated an extended model in which individual-level prior multiple infection probabilities are described with a logit linear predictor of putative risk factors. On simulated data, this model accurately estimated the true risk ratio associated with a covariate leading to a two-fold higher probability of harboring a multiple infection (risk ratio (RR) median [95% HPD] 1.74 [1.08–2.48]) in the context of four additional background null covariates (S8 Table.).

### 3.3. Prevalence of HIV multiple infections among sequenced participants

We next considered estimating the prevalence of multiple infection in the sequenced sample of 2,029 participants living with viremic HIV. In a model accounting for partial sequencing success and false-positive and false-negative observations of multiple subgraphs we estimate that 92 (4.53%) of the sequenced viremic PLHIV had a median posterior probability of multiple infection greater than 50% when allowing the probability of multiple infection ($\delta_i$) to vary by age, sex, and community type (Figs 3A, 3B, and S6). Our empirical analyses above demonstrated that the number of genome windows with multiple subgraphs is less than would be expected in the absence of false-negatives (Fig 1D). In line with this observation, the model estimated a high false-negative rate (median [95% HPD] 57.63% [53.27%–61.99%], S9 Table.), implying that empirical phylogenetic signatures of multiple infection under-estimate the true infection status of individuals in any single HIV genomic window. It was therefore essential to have whole-genome data from a subset of participants (Fig 1) to estimate false-negative detection rates. Further, informed by the 91.08% of samples with no multiple subgraph windows, we estimated the false-positive rate to be low (0.32% [0.26%–0.4%]). However, we note that even a low absolute rate will likely give rise to spurious multiple subgraph observations in a large sample size, which warrants consideration in our statistical framework.

In this model, the estimated prevalence of multiple infections in the study sample was 5.86% [4.65%–7.21%] (S9 Table.). Relaxing our minor subgraph frequency-based filtering step resulted in only a slightly higher prevalence of multiple infections in the study sample (6.1% [4.86%–7.39%], S10 Table.). When considering only genome windows spanning the

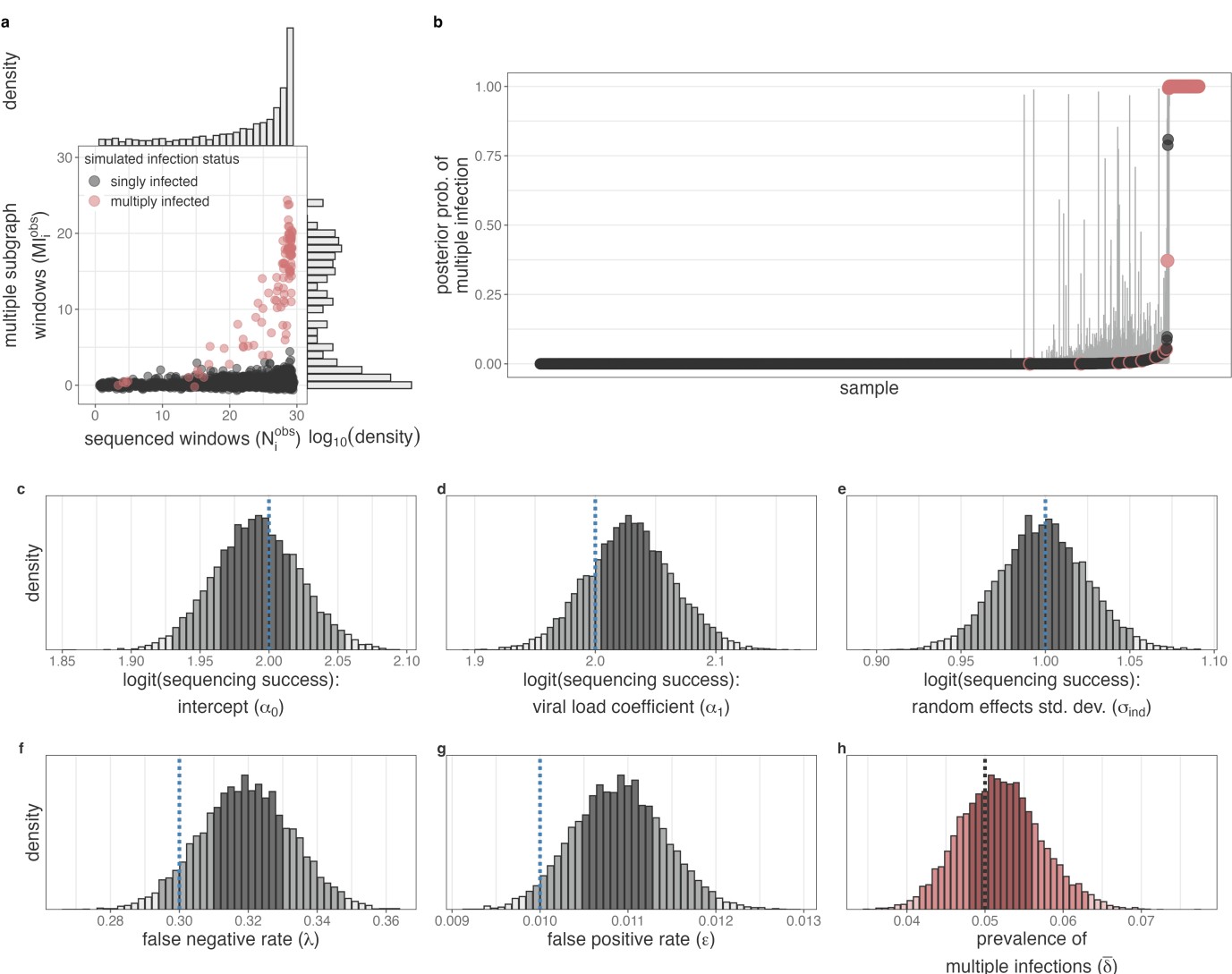

**Fig 2. Verification of model accuracy for estimating multiple infection prevalence on simulated data with incomplete sequencing success and false-negative and false-positive observations.** (A) Number of windows with sequence data (x-axis) v. number of windows with multiple subgraphs (y-axis) for each simulated sample. Data from multiply infected samples is highlighted in red. Marginal distributions are shown at right and above. (B) Estimated posterior probability of multiple infection for each sample. Confidence bounds represent the 95% highest posterior density. Data for each sample is shaded as in (A). (C-H) Posterior distributions of the baseline sequencing success ($\alpha_0$, C), dependence of sequencing success on viral load (log$_{10}$ copies/mL) standardized to mean = 0 and standard deviation = 1. ($\alpha_1$, D), standard deviation of per-individual sequencing success random effect ($\sigma_{ind}$, E), the multiple subgraph false-negative rate ($\lambda$, F), the multiple subgraph false-positive rate ($\epsilon$, G), and the population prevalence of multiple infections ($\bar{\delta}$, H). Posterior distributions in (C-H) bins are shaded according to the 95% and 50% HPD. Histogram bin width is calculated such that there are approximately 50 bins over the range of the plotted values. True values are shown as vertical dotted lines.

p24 and gp41 regions as in previous studies (e.g. [2,29,30]), we were unable to estimate $\sigma_{ind}$ with suitably high effective sample size (ESS) values as there were at most two regions of data for each sample. We therefore fixed $\sigma_{ind}$ = 0.7 based on the whole-genome analysis (S9 Table.) and found that that the sample prevalence of multiple infections based on p24 and gp41 was considerably lower as compared to the whole-genome analysis (2.31% [0.71%–4.94%], S11 Table.), highlighting the utility of incorporating whole-genome data into our inference.

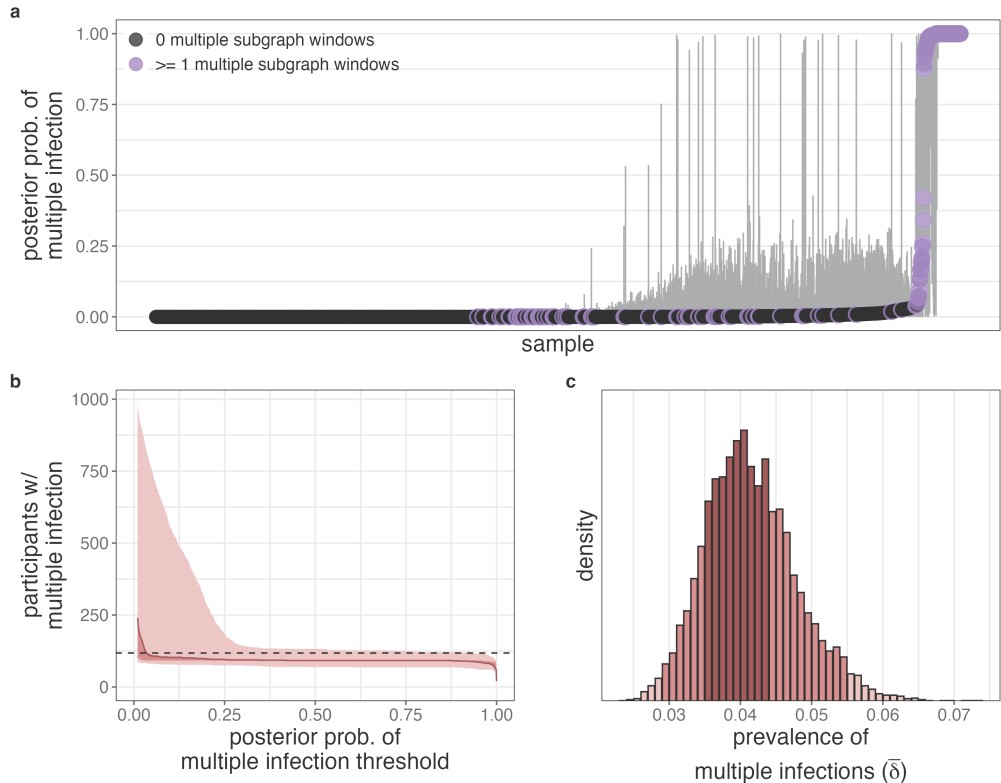

**Fig 3. Individual-level estimates and population-level characteristics of HIV multiple infection in people with viremic HIV in the Rakai Community Cohort Study, 2010-2020.** (A) Estimated posterior probability of multiple infection for each participant. Confidence bounds represent the 95% highest posterior density. Participants with at least one multiple subgraph window are shown in purple. (B) Number of participants with multiple infection as a function of the threshold used to dichotomize the probability of multiple infection. Central estimate uses the median estimated prevalence of multiple infections and shading uses 95% and 50% HPD. Horizontal dotted line plotted at the number of participants needed to match the estimated population prevalence of multiple infection. (C) Posterior distribution of the prevalence of multiple infections among viremic participants in the RCCS after accounting for sampling biases. Bins are shaded according to the 95% and 50% HPD. Histogram width is calculated such that there are approximately 50 bins over the range of the plotted values.

Finally, after adjusting for slight biases in the availability of sequence data among viremic participants (Table 1) using post-stratification based on age, sex, and community type ( S4 Table), the prevalence of multiple infections among viremic PLHIV in the RCCS was estimated to be slightly lower than the prevalence in the sequenced sample (4.09% [2.95%–5.45%], Fig 3C).

We next used our model to identify individuals with likely multiple infection based on their within-host phylogenetic trees and our modeling framework. Classification was based on the inferred, posterior multiple infection probabilities, and therefore our model-based approach accounted for individual-level factors associated with sequencing success and population-level false-postive and false-negative rates. We determined a binary classification cut-off above which individuals were classified as having a likely multiple infection such that the total number of identified individuals was consistent with the estimated prevalence in the sample, which resulted in a cut-off of 3.5%. Using this threshold, we estimated there were 118 individuals with a likely multiple infection (Fig 3B).

## 3.4. Risk factors of HIV multiple infection

In African contexts, HIV infection risk varies at the individual-level, such as by age, gender, sexual behaviour and circumcision status, and at the community-level [35,36,41]. We therefore next aimed to characterize individual and population-level risk factors for multiple infection with HIV. First, given the significantly higher prevalence of HIV and viremic HIV in Lake Victoria fishing communities [36,96], we investigated whether participants with viremic HIV in these communities had increased risk of multiple infection as compared to participants with viremic HIV in inland communities. Using the model described above with age, sex, and community type as predictors of the probability of multiple infection and accounting for sequencing biases through poststratificaiton we calculated the prevalence of multiple infections among viremic PLHIV in fishing and inland communities and found that multiple infections in fishing communities were 2.33 times (95% HPD 1.3–3.7)-times more frequent than in inland communities (with posterior median [95% HPD] prevalence of multiple infection of 7.42% [5.62%–9.31%]) and 3.14% [1.8%–4.74%] respectively, Fig 4A and S9 Table.). The estimated prevalence ratio for HIV multiple infection was therefore broadly comparable to the risk ratio of HIV prevalence and viremia in fishing as compared to inland communities (2.5-3)[36,96], consistent with the expectation that the risk of superinfection acquisition scales with the population prevalence of viremic HIV. Because participants from fishing communities are oversampled in our sequence data (Tables 1 and S4), this also explains the lower estimated prevalence of multiple infections in the population as compared to the sample.

We additionally incorporated a binary feature describing the sequencing technology used to generate the deep-sequence data from each participant to assess the extent of technical bias in our inferences. In a univariate analysis, we estimated that multiple infections were less common among participants sequenced using the bait-capture protocol (RR median [95% HPD]: 0.64 [0.4–0.94], S12 Table.). However, 50.45% of bait-capture sequenced participants were residents of fishing communities compared to 76.02% of amplicon sequenced participants. Consequently, in a bivariate model with community type, the estimated magnitude of the dependence of multiple infection status on sequencing technology was considerably reduced and no longer considered to be significant at the 95% level. (multivariate RR median [95% HPD] 0.77 [0.48–1.12], S13 Table.).

Participants with HIV in fishing communities also reported having more lifetime sex partners (S7 Fig), so we next assessed whether the risk of harboring a multiple infection differed by the number of self-reported lifetime sex partners within each of the two community locations. As women tend to under-report their number of sex partners relative to men [97], we restricted this analysis to male participants. The number of lifetime sexual partners generally increases with age, and so we standardized responses relative to the age-specific mean number of lifetime sexual partners among participants separately for the inland and fishing communities (S8 Fig). Among 997 male participants included in this analysis, 516 reported an exact number of lifetime sex partners, 477 responded they had three or more lifetime partners, and 4 did not provide a response. We imputed ambiguous responses and missing data within our inference framework by assuming responses were missing at random between people with and without multiple infection (Materials and methods).

In a bivariate model with community type and number of lifetime sexual partners we did not find a statistically significantly higher risk of multiple infection in male participants with more lifetime sexual partners in the context of substantial missing data and sampling over potential missing values using age-specific prior distributions. However, we note that the posterior effect size translated into an estimated more than two-fold higher risk of multiple infection between men living with viremic HIV in fishing communities associated with having 30

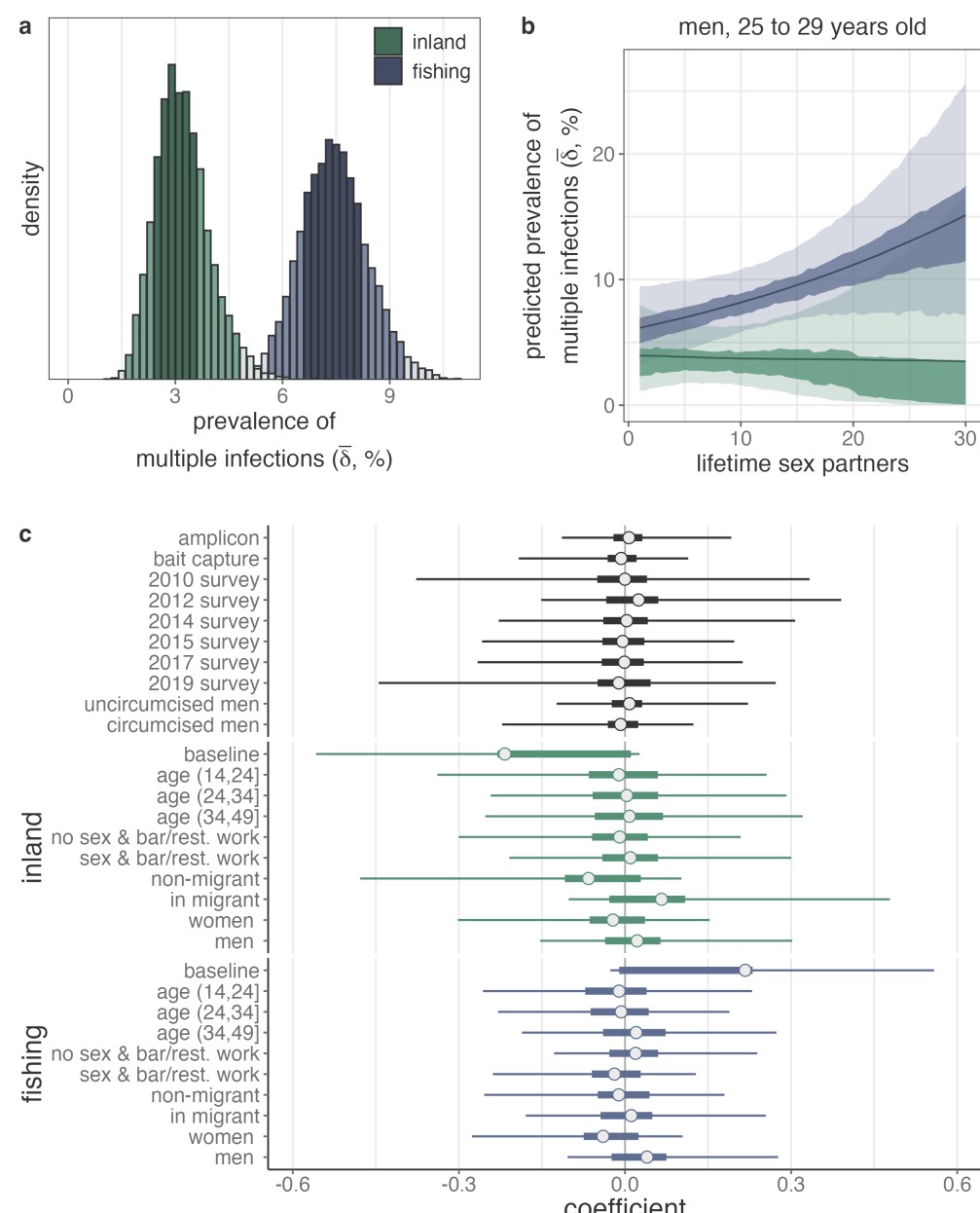

**Fig 4. Risk factors of HIV multiple infection among people with viremic HIV in the Rakai Community cohort Study, 2010-2020.** (A) Posterior distribution of the prevalence of multiple infections stratified by community type, accounting for sampling biases, estimated in a multivariate model (age, sex, and community type) with diffuse priors (*n* = 2,029). Bins are shaded according to the 95% and 50% highest posterior density (HPD). Histogram width is calculated such that there are approximately 50 bins over the range of plotted values. (B) Predicted risk of multiple infection among men aged 25 to 29 years old as a function of lifetime sex partners and community type estimated in a bivariate model with diffuse priors (*n* = 997). Median of the posterior distribution is plotted as the central estimate and shading represents the 95% and 50% HPD. Colors are as in (A). (C) Logistic coefficients for the association between putative risk factors and the probability of harboring a multiple infection estimated with Bayesian shrinkage priors (*n* = 1,970). Sex and bar/rest. work variable includes female sex and bar/restaurant worker and men who report having sex with female sex and bar/restaurant workers. Median of the posterior distribution is plotted as the central estimate, horizontal bars extend to the 95% and 50% HPD. Colors are as in (A).

lifetime sexual partners compared to one lifetime sexual partner (e.g. RR median [95% HPD] among 25-29 year olds 2.47 [0.7–5.61], Figs 4B and S9 for all age groups and S14 Table.). Very similar results were observed using a complete case analysis of the 516 men who provided an exact number of lifetime sex partners (S15 Table.).

We also performed a comprehensive discovery-based risk factor variable selection analysis over eight additive biological, behavioral and epidemic features, stratifying epidemiological and behavioral variables by community type to account demographic differences between the populations and excluding additional variable interactions. This analysis confirmed residency in fishing communities as a risk factor of multiple infection among sequenced participants, albeit with a wide credible interval, (multivariate RR median [95% HPD] 1.59 [0.92–2.85]), but did not identify any other variables that were associated with significantly higher or lower risk of multiple infection in our sample (Fig 4C and S16 Table.). Specifically, despite the fact that female bar/restaurant workers face a three-fold higher risk of incident HIV [41] we did not identify an increased risk of multiple infection among female bar/restaurant workers or men who have sex with bar/restaurant workers in either inland or fishing communities.

## 4. Discussion

In this large-scale study, we assessed the prevalence and risk factors of HIV multiple infection in an East African setting with high HIV burden using population-based pathogen deep-sequence surveillance data. To do this, we developed a Bayesian statistical model to identify multiple infections in deep-sequence phylogenies such as those generated by *phyloscanner* [38]. Our model incorporates false-negative and false-positive rates for the presence of genetically distinct viral variants and simultaneously estimates individual and population-level probabilities of harboring multiple infection. This framework also allows for the identification of biological and epidemiological risk factors for harboring a multiple infection. In simulation analyses, we demonstrated the ability of the model to generate accurate inferences across a range of parameter values, and fitted the model to *phyloscanner* within-host phylogenies inferred from HIV whole-genome RNA deep-sequence data collected between January 2010 and November 2020 from 2,029 viremic participants in the Rakai Community Cohort Study, a population-based open-cohort in southern Uganda. Among viremic participants in this study over the study period, the estimated prevalence of multiple infections was approximately 4%, reflecting the prevalence of co-circulating multiple infections present at time of sampling. Further, we showed that viremic participants with HIV living in high HIV prevalence fishing communities along Lake Victoria were more than twice as likely to harbor a multiple infection as compared to those living in inland agrarian or trading communities. Among male residents in fishing communities, we estimated that those with more lifetime sex partners can be expected to be more likely to have a multiple infection, although this finding did not reach statistical significance at the 95% level.

This study represents the largest analysis of HIV multiple infections by more than an order of magnitude [20] and rigorously accounts for partial sequencing success and uncertainty in individual-level estimates when estimating population-level risk of multiple infection. Our model indicated that in the context of incomplete genome coverage, as is common in HIV whole-genome sequencing [33], evidence for multiple infections is expected to be observed in only a subset of genome windows. However, we observed a high rate of false-negatives beyond what is expected due to incomplete sequencing, which may be due to insufficient diversity of infecting variants in some regions of the genome [95] to phylogentically distinguish them. This could potentially be due to recombination between infecting variants prior to sampling [4,5] such that infecting variants are only genetically distinct in some portions of genome

when sampled. The population-based multiple infection prevalence estimates from the data reported here are substantially more precise than previous estimates from this setting as expected given the larger sample size and slightly higher than previous estimates ($n$ = 7/149 [2]), likely primarily reflecting greater sensitivity of whole-genome sequencing data. Multiple infection among inland community study participants in this study (3.14%) was slightly less prevalent than in this earlier work (pre-2009, 4.7% [2]), consistent with reductions in HIV incidence over the same time frame [35]. Previous studies of female sex workers in urban Uganda and Kenya have estimated the prevalence of multiple infections to be as high as 14–16% in this high-risk demographic based on amplicon deep-sequencing [30,31]. Here, we do not replicate this finding using self-reported data on sex work or bar/restaurant work in our population-based sampling framework. We expect this is likely due to hesitation to self-report sex work among study participants and study participation bias among sex workers. However, our results are generally consistent with previous findings suggesting multiple infections are less common in African populations as compared to the United States (10–15% in studies conducted between 1996 and 2010 [98–102]), which may reflect the fact that the HIV epidemic in the United States is concentrated among men who have sex with men (MSM) and people who inject drugs (PWID) as opposed to the generalized nature of the epidemic in Africa. Further, as the risk of HIV transmission given exposure is 8–16× and 3–17× greater for needle-sharing and anal intercourse, respectively, as compared to vaginal intercourse [103], the risk of multiple infection acquisition given exposure may also be significantly greater in concentrated epidemics. To date, however, we note that the sample size of HIV multiple infection studies in the United States are relatively small (<150 individuals) and there is therefore significant uncertainty in the true underlying prevalence in these settings.

Our results add to considerable previous research on increased risk of HIV infection among Lake Victoria fishing communities. Previous studies have shown that overall HIV prevalence and prevalence of viremic HIV in these communities is 2.5–3× higher than in inland communities [36,96], in part due to migration of PLHIV to these communities [104, 105]. Further, despite a rapid increase in antiretroviral therapy (ART) uptake among residents of fishing communities over the study period [106], there remains a higher prevalence of people living with viremic, ART-resistant HIV as compared to inland communities [107]. We here show that viremic PLHIV in fishing communities also face a significantly higher burden of HIV multiple infections. We also show that among men in fishing communities, multiple infection risk increases with the number of lifetime sex partners. The precision of this estimate is hindered by a large proportion of qualitative responses to this component of the RCCS survey. These results imply that PLHIV in fishing communities continue to be exposed to viremic partners following initial infection. Public health interventions directed at viremic PLHIV in these communities may therefore not only provide life-saving treatment to these individuals but also reduce opportunities for the generation of novel recombinant forms of HIV which could pose challenges to control efforts through potential generation of more transmissible variants and broadening the antigneic space that potential vaccines need to cover [8–10,108–110].

We expect that our inferential framework may be adaptable to whole-genome deep-sequence phylogenies from other pathogens in which infection is chronic (thereby allowing sufficient time for superinfection to occur). Hepatitis C virus (HCV), which is a chronic viral infection transmitted either sexually or by injection drug use, is a natural extension [111]. Among people who inject drugs, the prevalence of HCV mixed infections is estimated to be as high as 39% [112]. Our framework has the advantage that it uses data from across the genome and does not require haplotyping of sequencing reads, which has proven to be exceedingly difficult with short-read sequence data [113]. Recent work has also attempted to

identify multiple infections of *Mycobacterium tuberculosis* (MTB), a chronic bacterial infection canonically of the lungs [114]. These methods work by either clustering allele frequencies to distinguish within- and between-variant differences [115–117] or by comparing sampled sequence data to a database of reference strains [118]. They therefore require defining circulating genetic diversity *a priori* (which may be challenging in a poorly sampled epidemic) or assume independence between alleles, failing to account for linkage between adjacent genome positions and the evolutionary history giving rise to the observed genetic variation. Multiple infections may also be of interest in acute, high-prevalence infectious diseases. For example, severe acute respiratory syndrome coronavirus 2 (SARS-CoV-2) superinfections have been observed by identifying mixed alleles as known lineage-defining sites [119,120].

While *deep-phyloMI* builds upon previous investigations into HIV multiple infections to provide more rigorous estimates of individual and population level parameters, we do rely on some simplifying assumptions in our framework. First, we only identify multiple infections among viremic participants with available deep-sequence data who were identified as part of a putative transmission network. While we adjust for known sampling biases based on demographic characteristics, there may be residual bias such that our sample is non-representative of the underlying population of viremic PLHIV. Further, because the RCCS did not perform viral load testing on all participants prior to the 2014 survey round we adjust only to the demographic characteristics of viremic PLHIV in the four most recent surveys. Further, we focus on identifying multiple infections only in cross-sectional sequence data. As multiple infections can be transient [31], we are unable to identify participants who have been but are not currently multiply infected. It is likely that longitudinal sampling or sequencing of the viral reservoir would identify additional individuals who have been multiply infected. Further, with only a single sample per-individual we were unable to reliably identify factors causally associated with incident multiple infection [121] and therefore report factors that are associated with prevalent multiple infections. Similarly, in the absence of longitudinal data or data sampled soon after initial infection we are unable to reliably distinguish multiple infections acquired through coinfection and superinfection. However, based on our parametrization of the *k* parameter within *phyloscanner* and the genetic distance between observed multiple subgraphs, we suspect that the vast majority of identified multiple infections are due to superinfection with a genetically distinct viral genotype. More liberal values of *k* would increase the sensitivity of our approach to identify closely related viral genotypes (such as those acquired during co-infection) at the expense of an increased rate of false-positives. Further, more liberal values of *k* would be appropriate in settings with less circulating HIV genetic diversity as compared to our study site [7].

HIV multiple infections complicate global control efforts by fueling the generation of genetic diversity [6], worsening clinical outcomes [15,16], and increasing viral load [16,31, 122]. Here we developed a robust inference framework to identify multiple infections in deep-sequence data and assess the role of epidemiological risk factors, such as living in high burden communities, in harboring multiple infections. This work will inform interventions aimed at preventing the acquisition of HIV superinfections and efforts to model the role iof multiple infections in the dynamics and evolution of HIV.

## Supporting information

**S1 Fig. Sequencing coverage among samples from 2,029 Rakai Community Cohort Study participant-visits contributed by viremic people living with HIV with $N_i^{obs} > 0$, stratified**

**by viral load category and sequencing technology.** (A) Distribution of $N_i^{obs}$ values for all samples. (B) Number of samples with coverage in each of the 29 genome window.
(TIF)

**S2 Fig. Pairwise genetic between unique tips in within-host phylogenetic trees among people with viremic HIV in the Rakai Community Cohort Study, 2010-2020.** Bins are shaded based on whether tips were assigned to the same subgraph (grey) or different subgraphs (purple), in the case where multiple subgraphs were observed.
(TIF)

**S3 Fig. Posterior distribution of parameters in full model fit to simulated data across a range of $\delta$ values.** Rows represent model to fit to simulated data with $\delta = 0$ (top row), $\delta = 5\%$ (second row), $\delta = 10\%$ (third row), and $\delta = 20\%$ (bottom row). Posterior distributions bins are shaded according to the 95% and 50% highest posterior density. Histogram width is calculated such that there are approximately 50 bins over the range of plotted values. True values are shown as vertical dotted lines. VL = viral load ($\log_{10}$ copies/mL) normalized to mean = 0 and std. dev = 1. Std. dev. = standard deviation.
(TIF)

**S4 Fig. Posterior distribution of parameters in full model fit to simulated data across a range of $\lambda$ values.** Rows represent model to fit to simulated data with $\lambda = 0.1$ (top row), $\lambda = 0.2\%$ (second row), $\lambda = 0.3\%$ (third row), and $\lambda = 0.4\%$ (bottom row). Posterior distributions bins are shaded according to the 95% and 50% higheset posterior density. Histogram bin width is calculated such that there are approximately 50 bins over the range of the plotted values. True values are shown as vertical dotted lines. VL = viral load ($\log_{10}$ copies/mL) normalized to mean = 0 and std. dev = 1. Std. dev. = standard deviation.
(TIF)

**S5 Fig. Posterior distribution of parameters in full model fit to simulated data across a range of $\epsilon$ values.** Rows represent model to fit to simulated data with $\epsilon = 0$ (top row), $\epsilon = 0.5\%$ (second row), $\epsilon = 1\%$ (third row), and $\epsilon = 5\%$ (bottom row). Posterior distributions bins are shaded according to the 95% and 50% higheset posterior density. Histogram bin width is calculated such that there are approximately 50 bins over the range of the plotted values. True values are shown as vertical dotted lines. VL = viral load ($\log_{10}$ copies/mL) normalized to mean = 0 and std. dev = 1. Std. dev. = standard deviation.
(TIF)

**S6 Fig. Individual-level estimate of HIV multiple infection in people living with viremic HIV in the Rakai Community Cohort Study, 2010-2020.** Estimated posterior $\log_{10}$ probability of multiple infection for each participant. Confidence bounds represent the 95% highest posterior density. Participants with at least one multiple subgraph window are shown in purple.
(TIF)

**S7 Fig. Mean number of lifetime sex partners stratified by HIV serostatus, sex, community type, and age among 109,608 RCCS participant-visits.** Excludes participant visits in which respondents provided a categorical response (N = 5,436 (10.67%)).
(TIF)

**S8 Fig. Standardization curve used to adjust observed number of lifetime sex partners among men for age-cohort effects.** Includes simple imputation of categorical responses (e.g. "1-2" and "3+") to 1) the mean value of observed responses of 1 or 2 ("1-2") within age

category and community type and 2) the mean of a lognormal distribution fit to observed responses of ≥ 3 lifetime sex partners within age category and community type.
(TIF)

**S9 Fig. Posterior estimates of the prevalence of multiple infections, stratified by age category and community type.** Median estimate is plotted as a line and shading represents the 50% and 95% highest posterior densities. All age categories share the same coefficient estimates but differ because lifetime sex partner values are standardized to the mean of the observed values within groups defined by sex, age category, and community type.
(TIF)

**S1 File. Supplementary methods.**
(PDF)

**S2 File. Reference genomes included in the *phyloscanner* analysis.**
(TXT)

**S3 File. Normalization constants used to adjust branch lengths in within-host phylogenetic trees.**
(CSV)

**S4 File. Sensitivity of results to choice of genome windows.**
(PDF)

**S1 Table. Count of participants sequenced using each sequencing protocol.**
(PDF)

**S2 Table. Characteristics of Rakai Community Cohort Study participant, 2010–2020.** For each participant, includes data from the participant-visit processed with PHSC if applicable or the participant-visit with the highest viral load, using the first visit in the case of ties or for people not living with HIV. Percentages represent the row percentages within each category. Binomial confidence intervals were calculated using the Agresti–Coull method. PHSC = phyloscanner.
(PDF)

**S3 Table. Count of missing values among 50,967 RCCS participants**. For each participant, includes data from the participant-visit processed with PHSC if applicable or the participant-visit with the highest viral load, using the first visit in the case of ties or for people not living with HIV. In each category the percentage represents the percentage of all participants or all participants that were viremic and processed with PHSC.
(PDF)

**S4 Table. Viremic participant-visits (2014–2019) and participants with available phyloscanner output belonging to epidemiological strata in the Rakai Community Cohort Study.** Epidemiological strata are defined by community type, age category, and sex. As viral load testing was not routinely conducted in earlier study rounds, the viremic participants belonging to each strata were tabulated using only data from the 2014 through 2019 surveys.
(PDF)

**S5 Table. Parameter estimates for base model fit to base simulated data.** ESS = effective sample size. HPD = highest posterior density.
(PDF)

**S6 Table. Parameter estimates for base model fit to full simulated data.** ESS = effective sample size. HPD = highest posterior density.
(PDF)

**S7 Table. Parameter estimates for full model fit to full simulated data.** ESS = effective sample size. HPD = highest posterior density.
(PDF)

**S8 Table. Parameter estimates for extended model fit to extended simulated data with epidemiological risk factor of multiple infection.** ESS = effective sample size. HPD = highest posterior density. stz-MVN = sum-to-zero multivariate Normal distribution.
(PDF)

**S9 Table. Parameter estimates for full model fit to deep-sequence data from 2,029 RCCS participants living with viremic HIV with age, sex, and community type as putative risk factors for harboring multiple infections** ESS = effective sample size. HPD = highest posterior density. stz-MVN = sum-to-zero multivariate Normal distribution.
(PDF)

**S10 Table. Parameter estimates for full model fit to deep-sequence data from 2,029 RCCS participants living with viremic HIV with age, sex, and community type as putative risk factors for harboring multiple infections.** Includes minor subgraphs supported in < 1% of reads in a given window so long as they are supported by at least three reads. ESS = effective sample size. HPD = highest posterior density. stz-MVN = sum-to-zero multivariate Normal distribution.
(PDF)

**S11 Table. Parameter estimates for full model fit to deep-sequence data from 1,742 RCCS participants living with viremic HIV with age, sex, and community type as putative risk factors for harboring multiple infections.** Includes data from genome windows spanning the p24 (1427–1816) and gp41 (7941–8264) regions. ESS = effective sample size. HPD = highest posterior density. stz-MVN = sum-to-zero multivariate Normal distribution.
(PDF)

**S12 Table. Parameter estimates for full model fit to deep-sequence data from 2,029 RCCS participants living with viremic HIV with deep-sequencing protocol as a putative risk factor for harboring multiple infections.** ESS = effective sample size. HPD = highest posterior density. stz-MVN = sum-to-zero multivariate Normal distribution.
(PDF)

**S13 Table. Parameter estimates for full model fit to deep-sequence data from 2,029 RCCS participants living with viremic HIV with community type and deep-sequencing protocol as putative risk factors for harboring multiple infections** ESS = effective sample size. HPD = highest posterior density. stz-MVN = sum-to-zero multivariate Normal distribution.
(PDF)

**S14 Table. Parameter estimates for full model fit to deep-sequence data from 997 men who participated in the RCCS living with viremic HIV with community type and number of lifetime sex partners as putative risk factors for harboring multiple infections adjusted for deep-sequencing protocol.** ESS = effective sample size. HPD = highest posterior density. stz-MVN = sum-to-zero multivariate Normal distribution.
(PDF)

**S15 Table. Parameter estimates for full model fit to deep-sequence data from 516 men who participated in the RCCS living with viremic HIV with community type and number of lifetime sex partners as putative risk factors for harboring multiple infections.** Excludes participants with ambiguous or missing data on the number of lifetime sex partners. ESS = effective sample size. HPD = highest posterior density. stz-MVN = sum-to-zero multivariate Normal distribution.
(PDF)

**S16 Table. Parameter estimates for full model fit to deep-sequence data from 1,970 RCCS participants living with viremic HIV with putative risk factors for harboring multiple infection and Bayesian shrinkage priors.** ESS = effective sample size. HPD = highest posterior density. stz-MVN = sum-to-zero multivariate Normal distribution.
(PDF)

## Acknowledgments

We thank the participants of the Rakai Community Cohort Study for making this research possible. Further, we thank all Rakai Health Sciences Program staff and all members of the PANGEA-HIV consortium. We thank Dr. Chris Wymant, PhD (Pandemic Sciences Institute, University of Oxford) for insightful discussions about Bayesian modeling of HIV multiple infections and helpful comments on this manuscript. We thank Zhi Ling (Saw Swee Hock School of Public Health, National University of Singapore) for advice on enforcing sum-to-zero constraints in the context or horseshoe-type shrinkage priors. Computational resources were provided through the Imperial College Research Computing Service and the Biomedical Research Computing Cluster at the University of Oxford.

## Author contributions

**Conceptualization:** Michael A Martin, Andrea Brizzi, Alexandra Blenkinsop, Christophe Fraser, M. Kate Grabowski, Oliver Ratmann.

**Data curation:** Michael A Martin, Andrea Brizzi, Xiaoyue Xi, Ronald Moses Galiwango, Sikhulile Moyo, David Bonsall, Gertrude Nakigozi, Godfrey Kigozi, M. Kate Grabowski, Oliver Ratmann.

**Formal analysis:** Michael A Martin, Andrea Brizzi, Xiaoyue Xi, Alexandra Blenkinsop, M. Kate Grabowski, Oliver Ratmann.

**Funding acquisition:** Sikhulile Moyo, Andrew D Redd, Lucie Abeler-Dörner, Christophe Fraser, Steven J Reynolds, Thomas C Quinn, Joseph Kagaayi, David Bonsall, David Serwadda, Gertrude Nakigozi, Godfrey Kigozi, M. Kate Grabowski, Oliver Ratmann.

**Investigation:** Michael A Martin, Ronald Moses Galiwango, Deogratius Ssemwanga, Andrew D Redd, Steven J Reynolds, Thomas C Quinn, Joseph Kagaayi, David Bonsall, Gertrude Nakigozi, Godfrey Kigozi, Oliver Ratmann.

**Methodology:** Michael A Martin, Andrea Brizzi, Xiaoyue Xi, Alexandra Blenkinsop, David Bonsall, Oliver Ratmann.

**Project administration:** Michael A Martin, Sikhulile Moyo, Lucie Abeler-Dörner, Christophe Fraser, Steven J Reynolds, Thomas C Quinn, Joseph Kagaayi, David Bonsall, David Serwadda, Gertrude Nakigozi, Godfrey Kigozi, M. Kate Grabowski, Oliver Ratmann.

**Resources:** Thomas C Quinn, Joseph Kagaayi, David Bonsall, M. Kate Grabowski, Oliver Ratmann.

**Software:** Michael A Martin, Oliver Ratmann.

**Supervision:** Christophe Fraser, Steven J Reynolds, Thomas C Quinn, Joseph Kagaayi, Gertrude Nakigozi, Godfrey Kigozi, M. Kate Grabowski, Oliver Ratmann.

**Validation:** Michael A Martin, M. Kate Grabowski, Oliver Ratmann.

**Visualization:** Michael A Martin.

**Writing – original draft:** Michael A Martin, M. Kate Grabowski, Oliver Ratmann.

**Writing – review & editing:** Michael A Martin, Andrea Brizzi, Xiaoyue Xi, Ronald Moses Galiwango, Sikhulile Moyo, Deogratius Ssemwanga, Alexandra Blenkinsop, Andrew D Redd, Lucie Abeler-Dörner, Christophe Fraser, Steven J Reynolds, Thomas C Quinn, Joseph Kagaayi, David Bonsall, David Serwadda, Gertrude Nakigozi, Godfrey Kigozi, M. Kate Grabowski, Oliver Ratmann.

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
