## [Decision Letter · Decision Letter 0]

17 Dec 2024

PPATHOGENS-D-24-02277

Quantifying prevalence and risk factors of HIV multiple infection in Uganda from population-based deep-sequence data

PLOS Pathogens

Dear Dr. Martin,

Thank you for submitting your manuscript to PLOS Pathogens. We apologise for the slow turn-around time, due to difficulty in obtaining reviewers. After careful consideration, we feel that it has merit but does not fully meet PLOS Pathogens's publication criteria as it currently stands. Therefore, we invite you to submit a revised version of the manuscript that addresses the points raised during the review process.

Please submit your revised manuscript within 60 days Feb 15 2025 11:59PM. If you will need more time than this to complete your revisions, please reply to this message or contact the journal office at plospathogens@plos.org. Please include the following items when submitting your revised manuscript:

We look forward to receiving your revised manuscript.

Kind regards,

Penny L. Moore

Academic Editor

PLOS Pathogens

Richard Koup

Section Editor

PLOS Pathogens

 Sumita Bhaduri-McIntosh

Editor-in-Chief

PLOS Pathogens

orcid.org/0000-0003-2946-9497

 Michael Malim

Editor-in-Chief

PLOS Pathogens

orcid.org/0000-0002-7699-2064

**Journal Requirements:**

At this stage, the following Authors/Authors require contributions: Michael A Martin, Andrea Brizzi, Xiaoyue Xi, Ronald Moses Galiwango, Sikhulile Moyo, Deogratius Ssemwanga, Alexandra Blenkinsop, Andrew D Redd, Lucie Abeler-Dörner, Christophe Fraser, Steven J Reynolds, Thomas C Quinn, Joseph Kagaayi, David Bonsall, David Serwadda, Gertrude Nakigozi, Godfrey Kigozi, M. Kate Grabowski, and Oliver Ratmann. Please ensure that the full contributions of each author are acknowledged in the "Add/Edit/Remove Authors" section of our submission form.

4) We notice that your supplementary Figures, and Tables are included in the manuscript file. Please remove them and upload them with the file type 'Supporting Information'. Please ensure that each Supporting Information file has a legend listed in the manuscript after the references list.

5) Please ensure that the funders and grant numbers match between the Financial Disclosure field and the Funding Information tab in your submission form. Note that the funders must be provided in the same order in both places as well.

**Reviewers' Comments:**

Reviewer's Responses to Questions

**Part I - Summary**

Reviewer #1: Martin and colleagues present a novel method to identify multiple infections in PLWH using whole-genome deep-sequencing data. They use this method to quantify the incidence of multiple infection and infer risk factors associated with it. Their analysis of data sampled from PLWH in Uganda demonstrates a positive effect of high community HIV-1 prevalence on multiple infection acquisition risk. Their Bayesian model is thoughtful to account for partial sequencing and expected false-negative/positive rates, as well as conditioning the prior on viral load and sequencing protocol, and appears to perform well on simulated data. This could be a useful method in light of the proliferation of NGS data for HIV-1. The manuscript is generally well written and the method clearly documented. - Eric Lewitus

Reviewer #2: The Rakai project investigators present here a new study of HIV superinfection. While the findings themselves follow on many similar studies over the years demonstrating superinfection, particularly in high prevalence areas and populations, this study applies newer methods and samples a larger population, making these estimates of superinfection prevalence likely more robust for the population understudy. Overall, the study presents what looks to be technical advance that may be useful for these types of studies going forward although I was not able to evaluate the computational aspects of the method.

It was a bit hard to read as the authors are quite negative regarding earlier work which is a shame given some of it really set the foundation for understanding HIV superinfection.

**Part II – Major Issues: Key Experiments Required for Acceptance**

Reviewer #1: There is one major source of confusion that makes the manuscript (and its implications) difficult to process. While the authors articulate the distinction between co-infection (when multiple founder variants are established at acquisition) and superinfection (when a secondary infection is acquired, typically months or years following the initial acquisition), they do not clearly address which they are detecting or indeed can detect with their method. They state in the Discussion that they cannot "distinguish multiple infections through coinfection and superinfection", even though differences between these are ostensible and the authors could venture to differentiate between them. If indeed they are aiming to detect both, then their estimate of a 6% rate of multiply infected individuals is very low. However, it seems they are knowingly detecting superinfections (lines 647-651, 750-752) and even, by setting k=0.067 and removing rare variants, excluding co-infections. Perhaps the project started agnostic to the type of multiple infection, but eventually the authors realized they were only detecting superinfections. Fair enough. But for a reader who might try to use this method, it is essential they know what they can detect with it.

Associated risk factors, diversification profiles, effects on nAb development, and rates of occurrence are not always aligned for co-infections and superinfections (some references supplied in itemized comments), so I think it is important the authors clarify this point. Firstly, I suggest the they expand their simulations to identify lower diversity boundaries on the limits of detection with their method and compare those boundaries to typical diversity estimates for co-infections. This can then be applied to their analysis by comparing simulated results to subgraph distances to the MRCA in empirical analysis. Secondly, in light of those simulations, I recommend the authors are more explicit about the types of multiple infections they are detecting and re-center their manuscript on the superinfection literature. A manuscript more attuned to this would be stronger, I think. Alternatively, by setting k above the average distance between multiple founder variants at, say, six months and removing rare variants the authors could simply state that they are intentionally excluding co-infections (or trying to) and focus on superinfections.

While in no way does this caveat undermine the authors' method or results, I think making a harder distinction between co- and super-infections will clarify the applicability of their method and the context of their results.

Reviewer #2: No experiments needed specifically although if they do not have data where they can compare their method to other methods, then Mmke clear that this method was not directly shown to be better at detecting superinfection than other methods - that is just an assumption.

Note the limitations of cross sectional studies for understanding correlates

**Part III – Minor Issues: Editorial and Data Presentation Modifications**

Reviewer #1: Additional comments are listed as they appear in the manuscript.

Line 6: references 3 & 4 refer to superinfection only

Line 11: co-infection and superinfection have also been shown to augment nAb development (e.g., for co-infection: Lewitus et al, PLoS Path, 2022; for superinfection: Powell et al., J Vir, 2010; Cortez et al., J Vir, 2011; Krebs et al., Immunity, 2019), although this is less consistent for superinfections.

Lines 47-49: would help to mention that Piantadosi et al's inability to identify superinfection in env was due to recombination (not because infecting envs were similar)

Line 58: please mention some of the methods that exist for identifying co-infections (e.g., Keele et al., PNAS, 2008; Dearlove et al., PLoS CompBio, 2021; Lewitus & Rolland, Virus Evolution, 2019).

Line 61: What does "phylogenetically likely" mean? Likely based on the phylogeny?

Line 67: introduce RCCS acronym

Lines 84-87: this mostly repeats the top of the paragraph

Line 198: Was 250 bp chosen because it was the minimum read length? If so, please state that. What is the expected effect of using sequencing protocols with longer read lengths (e.g., Nanopore or PacBio)?

Lines 199-201: Were just HVs removed? Based on Fig 1, it looks like most of gp120 is absent.

Lines 203-4: How were the RCCS sequences subtyped?

Lines 216-244: It would be helpful to have a schematic, at least as a supplement, for the protocol described here.

Lines 238-240: I was confused by how k was parameterized. Accounting for divergence between variants prior to the host becoming infected assumes a co-infection, right? Or is this divergence from a reference? Or divergence from a transmission pair? Some clarifying language would be helpful. I think I'm getting stuck on what divergence is being referred to "prior to a given host becoming infected". Also, please try to use acquisition rather than infected when appropriate.

Line 241: I think k=0.067 would exclude most co-infections, certainly if sampled during acute infection.

Lines 246-249: Rare variants can be common in co-infections (Yifan Li, Dynamics Conference, 2024). This step therefore may be discarding co-infections. I think that is fine, but the authors should be clear that they are deliberately excluding possible co-infections.

Line 435: How many sequences per individual?

Line 455-6: 11% and 19% of 29 windows is 3 and 5.5, respectively. Where do the 2 and 3 come from?

Line 462: How many individuals were missing gag, env, and nef?

Lines 467-469: This is an important result, but points to the confusion in not distinguishing between co-infections and superinfections. The observed rate of coinfection is 25% (see Baxter et al, Lancet Microbe, 2023). If accounting for partial sequencing success serves to more identify multiple infections missed by other methods (or bulk sequencing), then it doesn't make sense that the derived rate here is so low if the authors are considering co- and super-infections. Which is why I think it's critical they clarify what they're detecting.

Fig 1C: Is this averaged across gene windows?

Fig 1F: See my previous comment about gp120 being removed or just HVs?

Lines 483-485: If the authors are claiming to detect co-infecions as well, then I suggest they compare their method to existing ones (e.g., Keele et al., Gap Procedure, MGL).

Fig 2A: why the switch from purple to red in depicting multiply infected?

Fig 2C-H: How should we interpret the x axes?

Fig 2E,F: panel labels are incorrect in legend

Line 526: The authors could contextualize this in the Discussion in other estimates of superinfection (I think 3-8% is typically cited, but as the authors have shown this will no doubt vary by community)

Fig 3A: I don't think this is a clear presentation of these data. Perhaps the authors could ln-transform the posterior probabilities to show any small shifts in median values?

Line 602: Fig 4C reports logistic coefficient - how does this relate to RR median?

Lines 640-1: Do any gene windows (or combination of windows) perform as well as WGs?

Reviewer #2: The discussion mentions differences in prevalences of superinfection in this versus other studies but they dont discuss some important aspects- namely that people now are often treated and HIV spread in general is lower - both first and second infections as a result. In many African cohort studies, HIV incidence has been declining steadily since ART. Can they estimate exposure in this group versus the earlier groups studied in any way to address this?

Related to the point above, it is important to make clear the superinfection estimate is for this cohort in this time period. It is not applicable to other cohorts.

The authors make the point this method improves detection but although this seems plausible, that is just a hypothesis at this point. The true test of this would be to apply this method to samples from a previous study and this should be noted.

The authors should also discuss the limitations of a cross sectional study

Note: I can’t evaluate the model aspects of this so have no comments there. But this aspect needs to be evaluated by someone who can.

PLOS authors have the option to publish the peer review history of their article (what does this mean?). If published, this will include your full peer review and any attached files.

Reviewer #1: **Yes: **Eric Lewitus

Reviewer #2: No

**Figure resubmission:**
---

## [Decision Letter · Decision Letter 1]

21 Mar 2025

Dear Dr Martin,

We are pleased to inform you that your manuscript 'Quantifying prevalence and risk factors of HIV multiple infection in Uganda from population-based deep-sequence data' has been provisionally accepted for publication in PLOS Pathogens.

Best regards,

Penny L. Moore

Academic Editor

PLOS Pathogens

Richard Koup

Section Editor

PLOS Pathogens

Sumita Bhaduri-McIntosh

Editor-in-Chief

PLOS Pathogens

orcid.org/0000-0003-2946-9497

Michael Malim

Editor-in-Chief

PLOS Pathogens

orcid.org/0000-0002-7699-2064

Reviewer Comments (if any, and for reference):

Reviewer's Responses to Questions

**Part I - Summary**

Reviewer #1: The authors have addressed my concerns and I particularly appreciate the additional sensitivity analyses.

**Part II – Major Issues: Key Experiments Required for Acceptance**

Reviewer #1: (No Response)

**Part III – Minor Issues: Editorial and Data Presentation Modifications**

Reviewer #1: (No Response)

PLOS authors have the option to publish the peer review history of their article (what does this mean?). If published, this will include your full peer review and any attached files.

Reviewer #1: **Yes: **Eric Lewitus

---

## [Editor Report · Acceptance letter]

Dear Dr Martin,

We are delighted to inform you that your manuscript, "Quantifying prevalence and risk factors of HIV multiple infection in Uganda from population-based deep-sequence data," has been formally accepted for publication in PLOS Pathogens.

Best regards,

Sumita Bhaduri-McIntosh

Editor-in-Chief

PLOS Pathogens

orcid.org/0000-0003-2946-9497

Michael Malim

Editor-in-Chief

PLOS Pathogens

orcid.org/0000-0002-7699-2064